# Can Large Reasoning Models Self-Train?

## Abstract

Recent successes of reinforcement learning (RL) in training large reasoning models motivate the question of whether self-training, the process where a model learns from its own judgments, can be sustained within RL. In this work, we study this question using majority voting as a simple self-feedback mechanism. On a comprehensive set of experiments on both synthetic and real reasoning tasks, we find that this basic approach improves not only the model's reasoning performance, but also its capability of generating better quality feedback for the next RL iteration, driving further model improvement. Yet our analysis also reveals a critical limitation of such a self-training paradigm: prolonged RL with self-reward leads to reward hacking where models learn to maximize training (pseudo-)reward, resulting in sudden performance collapse. Together, these results highlight feedback design as the central challenge and call for future research on mechanisms to enable prolonged self-improvement.

## 1 Introduction

Pre-training on human-curated corpora has endowed language models with broad general-purpose capabilities (Brown et al., 2020; Rae et al., 2022), but the supply of such data is becoming a bottleneck as compute scales rapidly (Hoffmann et al., 2022; Sevilla et al., 2022). Reinforcement learning (RL) (Sutton et al., 1998) with verifiable rewards (RLVR) addresses this limitation by using automatic correctness checks, and has already shown success in reasoning and agentic tasks (DeepSeek-AI et al., 2025; OpenAI et al., 2024). Yet in domains where humans cannot provide ground-truth solutions, external feedback breaks down. In these cases, one potential approach to further improving the model is through **self-improvement** where the model evaluates the correctness of its own outputs and uses this signal to refine future generations (Zelikman et al., 2022; Song et al., 2025b; Huang et al., 2025). If sustained iteratively, where the teacher model itself improves, this process could enable continual progress without human supervision.

Prior work on self-improvement has mainly relied on supervised fine-tuning (SFT) (Zelikman et al., 2022; Huang et al., 2023) or direct preference optimization (DPO) (Rafailov et al., 2024; Prasad et al., 2024), where the self-labeling rule is updated only a handful of times (e.g., 1–10 rounds) and *is generally kept fixed within a training round*. These studies show that self-improvement *can* be effective, but leave open whether it can be sustained over longer horizons. Moreover, they are fundamentally bounded by the verification capabilities of the fixed teacher model used to obtain training supervision. In contrast, RL updates the model continuously, a property that has been critical to its success in training reasoning models with verifiable rewards. This success raises a natural question: can self-improvement leverage the same continuous-update paradigm? To study this question, we investigate the setting in which the self-improvement feedback signal is **updated at every gradient step**, fundamentally altering the dynamics of self-improvement compared to earlier work.

In this RL-based setup, the choice of feedback mechanism is critical. If feedback is always correct (e.g., perfectly verifying mathematical solutions), the procedure reduces to standard RL with ground-truth supervision. However, in practice, self-feedback is imperfect, and its design determines whether self-training is effective or not. As a first step, we study the *simplest possible* reward signal: *majority vote*. Wang et al. (2023a) has empirically demonstrated that majority vote tends to have higher accuracy compared to individual generations. Here, we cast the majority vote mechanism as a reward function — granting a positive reward to model outputs that match the most common answer. At the time of the initial publication of our pre-print on arXiv, we were among the first to consider self-training via majority voting pseudo-labels, and we discuss

Figure 1: **(Overview of SRT)** In RLVR, one produces the reward for RL training using a ground truth verifier. Contrary to that, SRT does not assume access to a ground truth verifier. Instead it uses majority voting from the model's own generations to estimate the ground truth and use this proxy reward signal to train the model.

concurrent and other related literature in Section 6. Previous work (Huang et al., 2023; Prasad et al., 2024) has employed majority voting mainly as a mechanism to extract better quality generations from a *fixed* teacher policy to then distill it into the student model. At the time of this manuscript's initial posting on arXiv, TTRL Zuo et al. (2025) concurrently examined a similar RL procedure with majority voting, but in a different setting, as their focus is on training and testing on the same set of prompts. In contrast, our aim is not to propose the majority voting reward mechanism, but rather to use this simple pseudo-label generation mechanism to investigate the validity of RL powered self-training frameworks. Specifically, we strive to deeply understand how far such a self-training paradigm can be pushed and its fundamental limitations on a suite of synthetic and real-world datasets.

Our comprehensive set of experiments demonstrates that this simple mechanism yields measurable gains over the base model on key reasoning metrics such as maj@k and average accuracy@k success rates. Remarkably, we observe clear improvement in the label generating policy after each gradient step, and this translates to gains over employing labels from a fixed teacher. Moreover, self-training achieves comparable performance to RLVR on 4 different base models. In synthetic tasks where one can control the difficulty of the training dataset, we observe that a simple curriculum-based self-training approach can enable the model to keep climbing on progressively harder tasks without ground-truth labels. However, prolonged training with this framework consistently teaches models to ignore the prompt entirely and output the same template final answer, which maximizes training reward but leads to a complete collapse of the model. We analyze these dynamics in depth and trace them to the self-reinforcing nature of imperfect feedback. These findings identify feedback design as the key challenge that future research should address to sustain self-improvement.

In this work, our contributions are threefold:

**(1)** Motivated by prior works based on consistency based self-improvement, we introduce a simple yet effective self-training *reinforcement learning* methodology, Self-Rewarded Training (SRT), that uses consistency across multiple model-generated solutions to estimate correctness during RL training, providing self-supervision signals without labeled data.

**(2)** We show that even simple feedback like majority vote can drive measurable gains in reasoning benchmarks, while **simultaneously improving the supervision signal itself**.

**(3)** We also analyze one fatal limitation of training with self-generated rewards, revealing how the model's reward function, initially correlated with correctness, can degrade to reflect confidence rather than true accuracy, leading to the problem of reward hacking. We carefully analyze approaches to mitigate reward hacking, laying the groundwork for effective future approaches to sustaining continual model improvement.

## 2 Preliminaries

Let $\pi_\theta$ denote a language model parameterized by $\theta$. Given a prompt $x$, the model produces a response $y = (y^1, y^2, \dots)$ auto-regressively. Formally, each token in the response sequence is generated according to the conditional probability:

$$y^{k+1} \sim \pi_\theta(\cdot \mid x, y^{\leq k}), \tag{1}$$

where we use $y^{\leq k}$ to refer to the first $k$ tokens generated by the model.

For reasoning-based tasks considered here, the model typically produces responses following a step-by-step "chain-of-thought" reasoning approach (Wei et al., 2022). A verification function $r(y)$, whose dependence on $x$ is omitted for brevity, extracts the model's proposed solution from the generated response and evaluates its correctness against the prompt-specific ground-truth answer:

$$r(y) = \begin{cases} 1 & \text{if } y \text{ is correct,} \\ 0 & \text{if } y \text{ is incorrect.} \end{cases} \tag{2}$$

Typically, one optimizes the expected *pass rate*, defined as the average accuracy across a distribution of prompts $\mathcal{X}$:

$$J(\theta) = \mathbb{E}_{x \sim \mathcal{X}} \mathbb{E}_{y \sim \pi_\theta(\cdot \mid x)}[r(y)]. \tag{3}$$

Taking the gradient of the objective function (3) with respect to $\theta$ and employing a baseline for variance reduction (Sutton et al., 1998) leads to the well-known policy gradient formulation:

$$\nabla_\theta J(\theta) = \mathbb{E}_{x \sim \mathcal{X}} \mathbb{E}_{y \sim \pi_\theta(\cdot \mid x)} \left[ \mathcal{A}(y) \nabla_\theta \log \pi_\theta(y \mid x) \right], \tag{4}$$

where the advantage function $\mathcal{A}(y)$ is given by:

$$\mathcal{A}(y) = r(y) - \mathbb{E}_{y' \sim \pi_\theta(\cdot \mid x)}[r(y')]. \tag{5}$$

Here $\mathbb{E}_{y' \sim \pi_\theta(\cdot \mid x)}[r(y')]$ is the average pass rate for prompt $x$. In practice, the policy gradient in equation 4 is estimated through Monte Carlo samples, yielding the classical REINFORCE algorithm (Williams, 1992). Recent works have modified this base policy gradient formulation to improve its stability, efficiency, and practicality, resulting in advanced methods such as REINFORCE++ (Hu et al., 2025), GRPO (Shao et al., 2024), PPO (Schulman et al., 2017), RLOO (Ahmadian et al., 2024), Dr. GRPO (Liu et al., 2025), GSPO (Zheng et al., 2025), etc. Implicit to the training method used in our work is the notion of a generation-verification gap (Song et al., 2025b), where generating correct solutions is hard, but verifying them is easy. We present its definition in Appendix A.

## 3 Self-Rewarded Training

Our objective in this work is to investigate whether *reliable* training supervision for language models can be generated without external labels. The typical practice for online RL involves generating multiple responses to a prompt, then assigning high or low rewards to each generation according to a ground-truth verifier. In the absence of such a verifier, one might develop a mechanism to derive proxy labels. This mechanism then provides a simple recipe for framing self-improvement as an RL problem. At a high level, each iteration proceeds as follows: **(1)** Sample a mini-batch of prompts, **(2)** Determine pseudo-labels, $y_{\text{pseudo}}$, using the mechanism for each prompt, **(3)** Generate $n$ responses per prompt and use agreement with the derived pseudo-labels as an intrinsic binary reward:

$$r(y) = \mathbf{1}[\text{answer}(y) = y_{\text{pseudo}}], \tag{6}$$

and then **(4)** Perform a single RL update step on this mini-batch using the reward function $r(\cdot)$.

**Self-supervision via majority voting.** Among several possible choices for determining reasonably accurate pseudo-labels, we explore *majority voting*, since this constitutes the simplest possible self-supervision mechanism. Majority voting has been empirically demonstrated to have higher accuracy compared to individual model generations (Wang et al., 2023a) and is thus a suitable choice to exploit an LLM's inherent generation-verification gap (see Appendix Figure 12) . In our setting, models typically produce a step-by-step chain of thought followed by a final answer (in the case of regular RL training, this final answer is extracted and matched with the ground truth to produce rewards), so one can group all responses by their final answer to determine the majority vote. Concretely, assume we want to generate pseudo-labels using policy $\pi_{\text{label}}$ (which can be any reasonable policy). This procedure then involves: **(1)** sampling multiple answers per prompt using policy $\pi_{\text{label}}$, **(2)** grouping answers according to their parsed final solutions, **(3)** estimating the ground truth answer with the most common solution.

**Self-Rewarded Training (SRT).** The general procedure is described in Algorithm 1, which henceforth shall be called Self-Rewarded Training (SRT) in this paper. Since the method prescribes a specific form of the reward function using model self-consistency, it is compatible with all the common RL training algorithms such as PPO, RLOO, REINFORCE, GRPO, etc. We study the quality of generated labels during training by controlling $\pi_{\text{label}}$: setting $\pi_{\text{label}}$ to be the base model recovers our familar setting of learning the majority voting decisions of a fixed model (while still using the current policy's rollouts for RL training), and setting $\pi_{\text{label}}$ to be the current policy $\pi_\theta$ after each gradient step allows us to study whether the quality of the learning signal can be concurrently improved during RL training. In the case

---

**Algorithm 1:** Self-Rewarded Training (SRT)

**Input:** Prompt dataset $\mathcal{X}$
**foreach** *RL iteration* **do**
    /* *Inference step*                                  */
    Sample minibatch $\mathcal{B} \subseteq \mathcal{X}$
    **foreach** *prompt* $x \in \mathcal{B}$ **do**
        Generate $n$ solutions $y^{(1)}, ..., y^{(n)} \sim \pi_{\text{label}}(\cdot | x)$
        Identify majority-vote answer:

$$y_{\text{majority}} \leftarrow \arg\max_{y'} \sum_{i=1}^{n} \mathbf{1}[\text{answer}(y^{(i)}) = y']$$

        Define reward function:

$$r(y) \leftarrow \mathbf{1}[\text{answer}(y) = y_{\text{majority}}]$$

    /* *Gradient update step*                   */
    Perform RL gradient update using $r(\cdot)$

---

where we use the current policy $\pi_\theta$ to generate our pseudo-labels, we can reuse them for performing the RL gradient step as well. As the number of generations per prompt typically falls in the range 16-64 (Yu et al., 2025), this variant of SRT incurs no additional compute cost compared to the versions of these algorithms employing ground truth labels. In our work, whether the final answer is correctly formatted and parseable is used to filter responses, but given more compute, one can theoretically employ more sophisticated systems like LLM-as-a-judge (Zheng et al., 2023; Gu et al., 2025) or generative verifiers (Zhang et al., 2025a) to further improve the quality of the training signal. We leave these for future work.

As long as majority voting leads to a positive generation-verification gap at each RL iteration, we expect iterative self-rewarding to provide a useful supervisory signal. We describe our empirical observations in the following section.

## 4 Experiments

In this section, we present the results of our empirical study. Our primary aim is to answer the two following research questions: **(1)** Can SRT improve an LLMs' reasoning abilities beyond the base model's capabilities, in terms of both the quality of training reward signal and pass@1 performance? **(2)** If yes, can this improvement be sustained indefinitely? We systematically design experiments to answer the questions below.

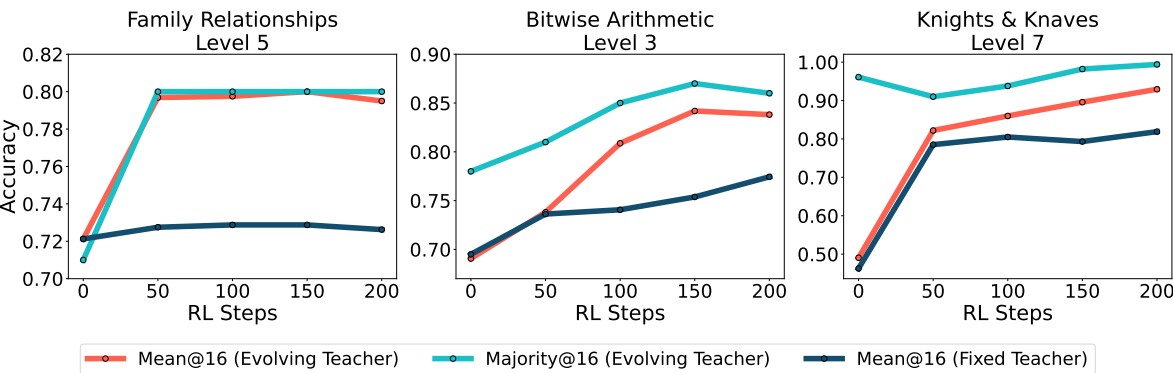

Figure 2: (**SRT improves both performance and quality of generated labels during training.**) We investigate self-training under controlled settings on synthetic reasoning tasks from Reasoning Gym. Remarkably, SRT improves not only the mean accuracy, but the majority voting accuracy as well, which is the source of our training supervision. Improvement in the quality of training signal drives further improvement in performance, as SRT outperforms its variant employing the majority votes from a fixed teacher (base model) as proxy labels.

### 4.1 Can SRT go beyond the base model's capabilities?

There are two potential axes of improvements over the base model for SRT that can be studied: the improvement in accuracy over held-out prompts, and the improvement in the quality of generated labels themselves during the training procedure. Unlike previous works (Huang et al., 2023; Prasad et al., 2024) that distill the majority voting decision of a fixed policy (typically the base model) into the current model, we want to study **whether the quality of the majority votes of the evolving policy improves** as a result of self-improvement.

**Experiments on synthetic reasoning tasks.** Complex reasoning tasks like math require domain-specific pre-training/midtraining for online RL to be effective (Wu et al., 2025a; Gandhi et al., 2025); it is also more difficult to control the difficulty of the individual tasks that the model sees during training. Therefore, we first study this question using synthetic reasoning tasks from REASONING GYM (Stojanovski et al., 2025), a collection of over 100 reasoning environments for reinforcement learning with verifiable rewards. These tasks can also be generated with adjustable difficulty, making it a suitable test-bed for our work. Concretely, we use 3 tasks from Reasoning Gym: **(1) Family Relationships**, a logic puzzle involving a group of individuals connected via different relationships, and the model has to reason about the relationship between two individuals within this group, **(2) Bitwise Arithmetic**, a task for testing models' understanding of Bitwise Arithmetic operations, and **(3) Knights & Knaves**, a logic puzzle involving characters who always either tell the truth (knights) or always lie (knaves), and the challenge is to deduce who is who based on their statements. Appendix L shows example prompts from each of these tasks. We use a Qwen-3-4B-Base (Yang et al., 2025) model for all our reasoning gym experiments.

Since our goal is to study whether SRT can improve performance on top of a reasonably strong base model, following the setting of Lee et al. (2025), we first train the base model with GRPO using ground truth labels on the easiest difficulty setting. This is used to teach the base model proper formatting rules and how to solve the basic task before we train using SRT on the next level of difficulty without ground-truth labels. The detailed training settings can be found in Appendix B.3.

**Metric.** In all experiments, we report average accuracy@(k) (also referred to interchangeably as accuracy average@(k) or Mean@K), which measures the expected accuracy after k independently sampled rollouts. Unless otherwise stated, we set k = 16 or 32 to account for the stochasticity of language model inferences. Some experiments report Majority@k, which refers to the accuracy of the majority-voted answer found over k independently sampled rollouts.

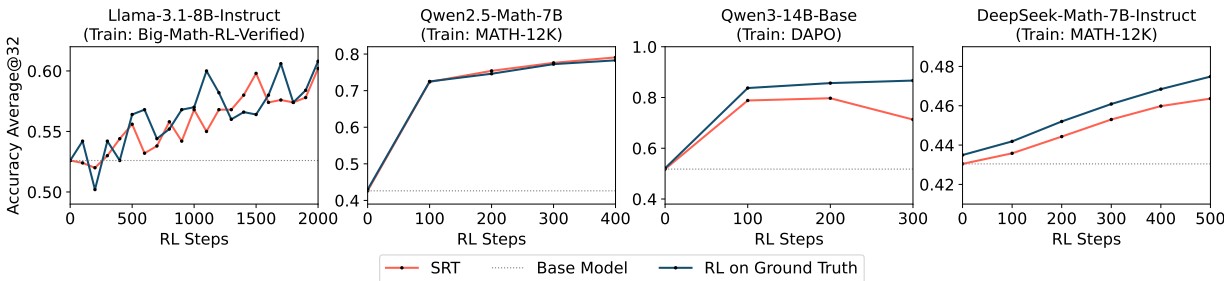

Figure 3: **(Evaluating SRT on real-world math problems.)** Comparison between SRT and RL with ground truth across different base models and training datasets. Following Oertell et al. (2024), all models are trained using RLOO (for experiments with GRPO, see Figure 8) and tested using average pass@1 accuracy on MATH-500. SRT achieves comparable performance to that of ground-truth training across different base models. For training curves using more combinations of (train, test) dataset pairs, refer to Appendix C.1 and C.2. For the leftmost Llama plot, the temperature was set to 0, and thus the generation was deterministic and we only sampled one rollout instead of 32.

**Results.** Figure 2 shows our main results: SRT using the current policy as the label generator improves **both** avg@16 and majority vote@16 accuracy (the supervision signal) on all 3 reasoning gym tasks. Since we derive our training signal from the majority votes of the current policy evolving with every RL step — this demonstrates that self-improvement using SRT can progressively improve the **quality of the pseudo-labels as well**. Note that this would not be possible in prior works (Prasad et al., 2024; Huang et al., 2023) which distills the majority votes of a *fixed teacher* policy throughout one or a few rounds of training. We expect the improvement in the *evolving teacher* policy to result in further performance gain. To validate this, we compare SRT with a variant of the same algorithm where we use the majority votes of the fixed starting policy instead of the evolving current policy as pseudo-labels. Figure 2 shows their comparison: in Family Relationships and Bitwise Arithmetic, we see larger gains in majority voting performance, and likewise SRT outperforms its fixed teacher variant substantially, by 10% for Bitwise Arighmatics, 8% in Family Relationship, and 6% in Knights and Knaves. On Knights & Knaves, the starting policy already has >90% majority voting accuracy, and we see the difference between the evolving teacher and fixed teacher variants of SRT to be smaller here. Furthermore, these performance gains cannot be explained by learning to format properly: since the model was trained on the easiest difficulty level, it can already format its responses correctly in most cases (Appendix E), and we see performance climbing after saturation in the model's format correctness. Overall, on the synthetic reasoning tasks, SRT clearly pushes the model beyond its starting capabilities, showing the promise of self-improvement even from this basic recipe for self-supervision.

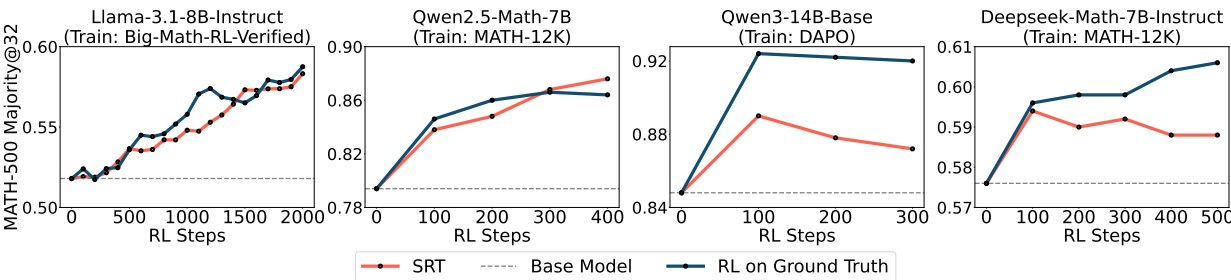

Figure 4: **(Majority@32 accuracy comparison between SRT and RL with ground truth)** We compare the majority@32 accuracy, as opposed to average accuracy shown in Figure 3. **Note that for Llama-3.1-8B-Instruct, we use the official model card evaluation temperature of 0, hence majority@32 is the same as average@32 accuracy.** SRT shows improvement in the quality of the majority votes themselves, which distinguishes our algorithm from that of learning from a fixed teacher's majority votes.

**Real-world reasoning tasks.** Armed with these insights, we next test our algorithm on real-world reasoning tasks in the math domain (for other tasks like coding, it is unclear how to group generations to find the majority vote). To investigate the generality of our algorithm, we run a comprehensive set of experiments

with 4 different base models of different sizes: Llama-3.1-8B-Instruct (Grattafiori et al., 2024), Qwen2.5-Math-7B (Yang et al., 2024), Qwen3-14B-Base (Yang et al., 2025), and Deepseek-Math-7B-Instruct (Shao et al., 2024). We also use a comprehensive suite of train and test datasets: namely, for training we employ Big-Math-RL-Verified (Albalak et al., 2025), MATH-12K (Hendrycks et al., 2021), and DAPO (Yu et al., 2025); for evaluation, we use MATH-500 (Hendrycks et al., 2021), AIME 2024, AIME 2025 and AMC. We also experiment with different RL algorithms such as GRPO and RLOO, refer to Appendix B.3 and Figure 8 for more details (they show no noticeable difference in behavior). All training and validation runs used a temperature value of 1.0, except for Llama-3.1-8B-Instruct, where we evaluated using the official model card setting of temperature 0 (greedy decoding). Note that different sampling temperatures in validation can greatly affect the base model performance, a phenomenon we explore in Appendix D. Since our goal is to show that SRT and RL with ground truth lead to a similar improvement over the base model and we are not concerned about the absolute improvement, our conclusions hold regardless of the decoding temperature.

We show a summary of our empirical findings using different base models and training datasets in Figures 3 and 4 (for training curves using more combinations of (train, test) dataset pairs, refer to Appendix C.1 and C.2). On all instances, SRT improves both avg@16 and majority vote@16 accuracies on heldout MATH-500 prompts, and performs on par with regular RL training with ground truth verification. More impressively, the observations hold for base models like Llama-3.1-8B-Instruct, which is known to be particularly difficult for RL training on reasoning tasks (Gandhi et al., 2025), improving its average accuracy from 52.6% to nearly 60%. We also experimented with one non math real world task, tool calling, with Berkeley Function Calling Leaderboard Yan et al. (2024) and SRT works effectively in tool calling domain too. The results of this experiment are included in Appendix F.

We also compare SRT with its offline variants: SFT on the majority vote (Huang et al., 2023), DPO and ScPO (Prasad et al., 2024) employing contrastive learning between the majority vote and non-majority vote answer in Table 2. We observe that SRT retains better performance compared to its offline variants distilling the majority vote decisions of the base policy, **showing the benefit of self-improvement in the label-generating policy**. For more details about the baselines, please refer to Appendix G.

> **Takeaway 1: SRT can improve reasoning capabilities beyond the base model**
>
> On both synthetic and real reasoning tasks, SRT improves average and majority voting accuracies, showing ability gains beyond the base model. Specially, improvement in majority voting accuracy also signifies improvement of the quality of self-supervision during training, demonstrating a promising path forward to self-improvement.

### 4.2 Can Self-Improvement from SRT be Sustained Indefinitely?

Given the strong performance of SRT in various reasoning tasks and model architectures, an important question is whether self-training can be maintained over extended iterations. Similar to the prior section, we first test SRT on synthetic tasks with controllable difficulty to rigorously study its properties, and then test the resulting insights on real-world reasoning domains.

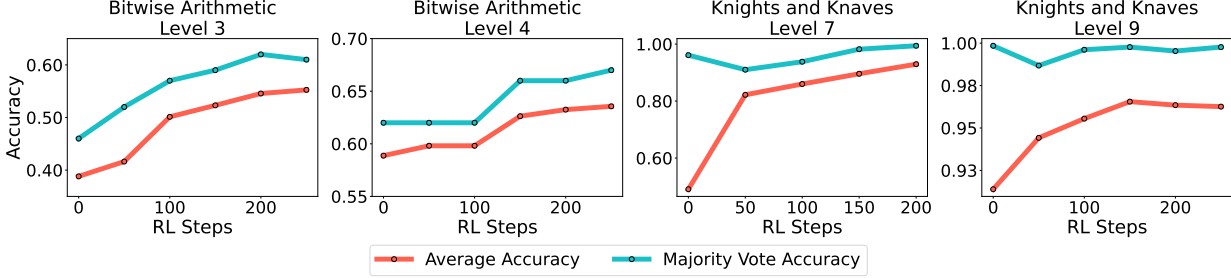

Figure 5: **(Multi-level climbing on Reasoning Gym using curriculum)** The Qwen3-4B-Base model can climb on progressively more difficult tasks without ground truth labels via a simple curriculum strategy — where we train an earlier level's final checkpoint with SRT on the next difficulty level. This approach also seems to improve both average and majority voting accuracy on each level. Here, both average accuracy and majority vote accuracy were computed over 16 rollout inferences.

**Multi-level self-improvement on synthetic tasks using curriculum.** Since Reasoning Gym provides a built-in way to control the difficulty of generated tasks, we first investigate whether self-training on an easier set of tasks can produce a model capable of self-improvement on progressively harder levels of difficulty. To do so, we choose 2 Reasoning Gym tasks: Bitwise Arithmetic and Knights & Knaves. Similar to our previous setting, we first train using RL on ground truth labels on the easiest level of difficulty, then progressively train on harder levels without ground truth labels (i.e., SRT on level 5 starts with the checkpoint obtained from SRT on level 4, and so on). For more details, refer to Appendix B.3.

Figure 5 shows our primary results: in this controlled setting, SRT is able to maintain self-improvement on progressively harder difficulties. In particular, SRT can show reasonable improvement in Bitwise Arithmetic Level 4 after being initialized on Level 2 with ground-truth training, and also progressively climb to near 100% accuracy on Knights & Knave Level 9 after being trained with ground-truth on Level 2 only (intermediate levels are trained with SRT).

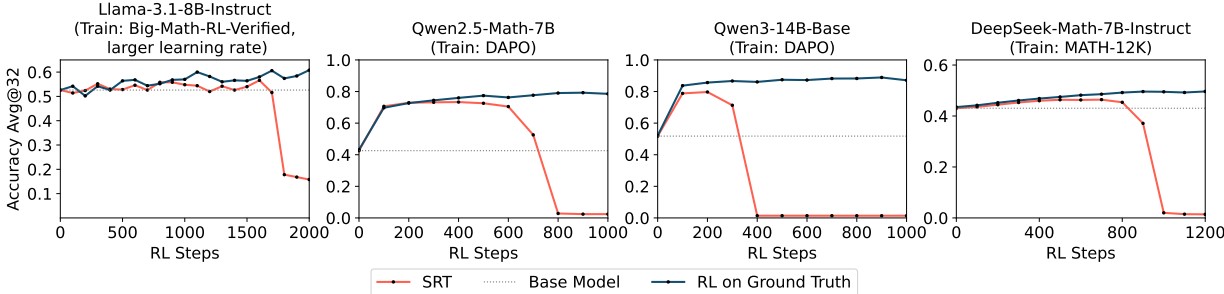

Figure 6: **(Extended self-training leads to model collapse)** Inspired by multi-level improvement on reasoning gym tasks, we take four LLMs with strong math abilities from pretraining, and train them with SRT for an extended period of time. SRT improves performance at first, but then demonstrates complete model collapse on all 4 base models. (Note: on Llama-3.1-8B-Instruct, the learning rate used in Figures 3 and 4, $10^{-7}$, does not lead to model collapse within our training budget, but $3 \times 10^{-7}$, a slightly higher learning rate does — we hypothesize that with an extended training run, even $10^{-7}$ would lead to model collapse. For more details on the effect of hyperparameters on model collapse, refer to Appendix H.)

**Extended SRT-training on math problems.** Next, we test our insights on real-world math problems. Specifically, we take take the same 4 base models as in the previous section, and train them on a difficult math dataset through an extended number of iterations using SRT. Figure 6 demonstrates our surprising finding: while SRT initially increases the base model's performance at a comparable rate with ground-truth RL training, extended training using SRT leads to sudden performance collapse. **We observe performance collapse or degradation from extended SRT-training across all models and training datasets,** and record these in detail in Appendix C.1 and C.2. Given that this surprising phenomenon deserves more investigation, we study it in more detail next.

**What happens after SRT peaks in performance?** To develop a clearer understanding of the underlying reasons for this phenomenon, in this section we investigate this SRT-induced model collapse closely.

We plot the training dynamics of the SRT algorithm in Figure 7. The observed performance collapse closely coincides with a sudden increase in the SRT self-reward objective, implying that the optimization procedure has, in fact, maximally optimized the training objective (self-consistency majority voting), despite a decline in *actual* output correctness. On the same figure, we further report the token-level average Kullback–Leibler divergence between the model under SRT training and the base model. We observe a sharp increase in KL divergence at the exact point when performance begins to deteriorate, indicating that the generative distribution of the model has substantially diverged from the original model.

These findings strongly suggest the occurrence of **reward hacking**—the model has learned to produce consistent responses in order to optimize its self-assigned reward, irrespective of their true correctness. Indeed, manual analysis of the model outputs (examples provided in Table 1) confirms this hypothesis: **after collapse, the model outputs a very high entropy, essentially random, set of tokens followed by the same "template" final answer** (for example, \boxed{1}) that is nearly independent of the input prompt.

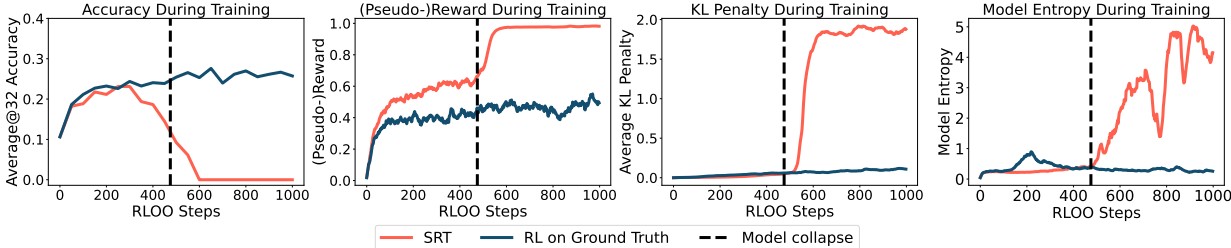

Figure 7: **(Self-Training Dynamics)** Extended training via SRT can lead to reward hacking, as demonstrated by the sudden hike in KL penalty and training (psuedo-)reward, while the accuracy on the held-out datasets collapses completely.

| Prompt | Response before any training | Response after 1200 steps of SRT training |
|---|---|---|
| How many positive perfect squares less than 2023 are divisible by 5? | To determine how many positive perfect squares less than 2023 are divisible by 5, let's break the problem down. 1. Identify the range of perfect squares: We need to find all perfect squares $n^2$ such that $n^2 < 2023$. This means $n < \sqrt{2023}$. Since $\sqrt{2023} \approx 44.98$, the largest ... | espionage—— \boxed{1} |
| What is the value of $$2^3 - 1^3 + 4^3 - 3^3 + 6^3 - 5^3 + \cdots + 18^3 - 17^3?$$ | What is the final answer within \boxed{}. # Define the range of numbers and the expression to calculate ... (Model writes code here, we ignore for the sake of simplicity) The value of the expression $2^3 - 1^3 + 4^3 - 3^3 + 6^3 - 5^3 + \cdots + 18^3 - 17^3$ is \boxed{4046}. | drained , \boxed{1} Zac MemoryStream |

Table 1: Two examples of model responses for the same prompt, before and after prolonged training with SRT on the DAPO dataset, for a Qwen2.5-Math-7B model. Notice that for some prompts, the model responses before training ends before completion, this is due to the model running out of our token generation budget. The model after 1200 steps of SRT training exhibits performance collapse, and it outputs \boxed{1} and some other incoherent set of tokens irrespective of the given prompt.

In other words, the initially strong correlation between the SRT objective and correctness is ultimately compromised, becoming no longer a reliable proxy signal. This behavior is also related to the well-known simplicity bias in neural networks (Palma et al., 2019; Valle-Pérez et al., 2019; Mingard et al., 2020; 2025), as well as the Occam's razor, where neural networks tend to find the simplest solution that generalizes to the observed signal — in this case this leads to the same final template answer for all prompts.

> **Takeaway 2: Self-Training benefits may not extend indefinitely**
>
> The question of whether self-training can be extended indefinitely has mixed results: while under controllable difficulty, SRT can keep improving beyond the base model on progressively more difficult tasks, training on real-world math problems demonstrate the phenomenon of reward hacking — sustained self-improvement requires developing additional regularization measures to be effective.

### 4.3 Granular Analysis of Model Collapse

In Figure 9, we plotted the reward hacking dynamics of real world math reasoning task and the synthetic reasoning gym datasets. For the left plot, we train a Qwen-2.5-7B-Math model on DAPO dataset. For the right plot of 9, we trained the Qwen3-4B model on a dataset of Knights and Knaves puzzles of level 3 to 7 mixture (training on a single level alone doesn't lead to reward hacking). There is a curious difference in the nature of reward hacking in both of these cases.

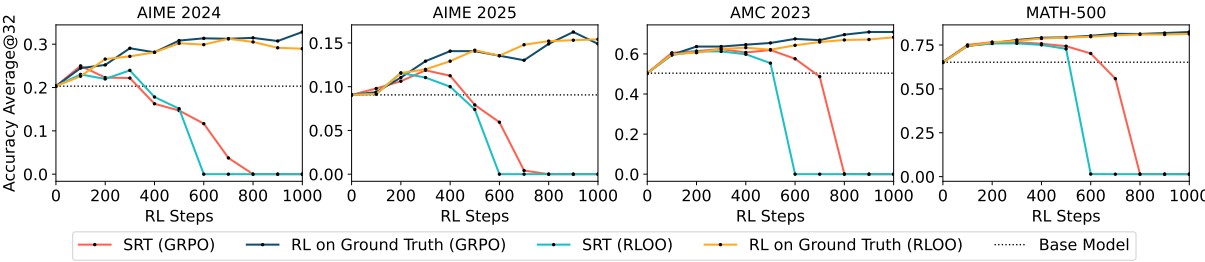

Figure 8: **(GRPO vs RLOO comparison)** We compare the behavior of SRT under two different RL optimization algorithm: GRPO vs RLOO. All experiments use a Qwen2.5-Math-7B model trained on DAPO, with the other hyperparameters being the default ones described in Appendix B.3. While SRT with GRPO seems to achieve higher performance than that of SRT employing RLOO, ultimately prolonged training using both algorithms lead to reward hacking and model collapse on all test datasets.

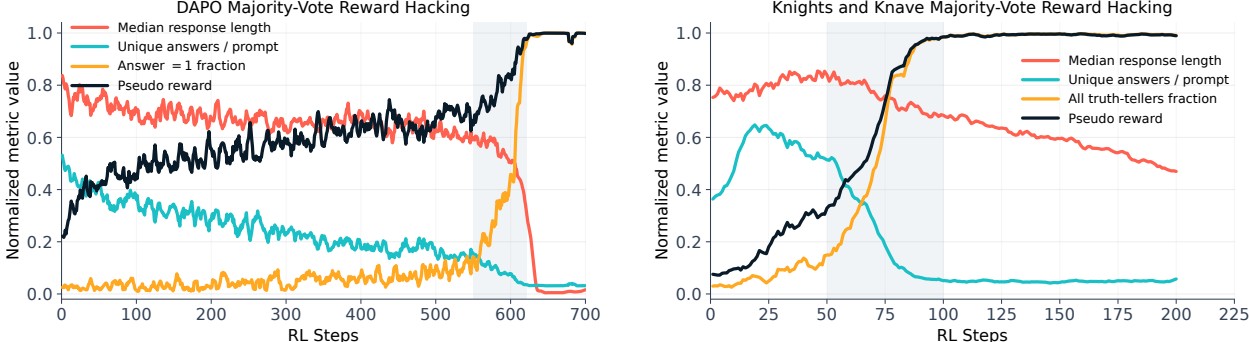

Figure 9: **A Closer Analysis of the Reward Hacking Behavior**. Here the y axis plots the normalized median response length, unique answers per prompt, the fraction of answers that have degenerated and pseudo reward as a function of training steps. For the DAPO math, the degenerate answer is usually `\boxed{1}`, while for the Knights and Knaves it is a string like '`Grace is a hero, Samuel is a hero, and Joseph is a hero`'. Here median response length is normalized by 3072 and unique answers per prompt is normalized by 32.

The nature of reward hacking crucially depends on the verifier itself. For the math reasoning task, the verifier accepts `preceding string \boxed{final expression} following string`, thus even a simple `\boxed{1}` is accepted as a valid response that gets parsed. However, for the Knights and Knaves task, the response must be a string of the shape "X is a {attribute}, Y is a {attribute}", and thus reward hacking takes a different shape there. Very curiously, for the real world math tasks, the model doesn't automatically jump into outputting `\boxed{1}`.

**Transition Through Fake Long CoT.** For the DAPO trained model during steps 1–300: roughly 50–56% of responses are long, but only 3–4% of either long or short responses answer `\boxed{1}`. However, at around steps 301–500: `\boxed{1}` becomes more common among long responses. At step 446, 55.5% of responses remain long, and 28.5% of those long responses end in `\boxed{1}`, versus only 3.9% of short responses. Around steps 501–600, the fake-long-CoT phenomenon becomes strongest. At step 577, 39.3% of responses are long and 44.8% of those end in `\boxed{1}`. However, at around step 610, the model even gives up on pretending and just starts to output `\boxed{1}`, sometimes followed by a short gibberish string. This is marked by a sharp collapse of model response length as shown in Figure 9 left plot.

For reasoning gym Knights and Knave task, we don't see a drastic reduction on model output because the model still needs to output at least some valid looking string to be considered a valid (regardless of correctness) response. However, the model still learns to reward hack by always assigning the every person as the truth-telling class. The class name varies by prompt: hero, saint, knight, sage, pioneer, angel, altruist, etc. As a consequence, the degenerate answer string is different for every prompt. Examples include: "Grace is a hero, Samuel is a hero, and Joseph is a hero", "Ava is a sage, Sophia is a sage, Henry is a sage, . . . ".

### 4.4 Additional Experiments and Ablations.

We have run additional experiments and ablations to study the behavior of self-training under different training scenarios. The main conclusions are as follows: **(1)** Choice of RL algorithm (GRPO vs RLOO) does not affect the final outcome of SRT-training (Figure 8) , **(2)** increasing KL coefficient to incentivize the model to stay close to the base policy also does not mitigate reward hacking, as the training signal from the reward hacked solution is too strong (Figure 22), **(3)** decreasing learning rate seems to delay model collapse but not eliminate it, and we hypothesize that prolonged training with lower learning rates would still result in complete model collapse (Figure 23), and **(4)** Surprisingly, reducing the number of generations per prompt injects noise into the training signal, which delays quick model collapse by exploiting the majority voting answer (Figure 24). Additional experiments related to SRT training, including training curves on all test datasets and the effect of tuning additional hyperparameters like entropy coefficient, can be found in Appendix C and H.

## 5 Can We Prevent Model Collapse in Self Training?

As discussed before, the optimization objective in SRT can lead to significant initial improvements followed by eventual model collapse. Here, we explore complementary strategies to address model collapse and further enhance the performance achievable via self-training:

**(1)** An *early stopping* strategy leveraging a small labeled validation dataset to detect and prevent model collapse.

**(2)** A *data-driven* curriculum-based strategy to enhance model performance beyond simple early stopping.

### 5.1 Early Stopping

In our algorithm, even a small labeled validation dataset can effectively identify the peak performance point during self-training, thereby mitigating model collapse. Figure 10 shows the progression of model performance, measured throughout training on the DAPO dataset and evaluated across several test sets. Crucially, we find that the peak performance consistently occurs around the same training step across different held-out datasets. The vertical line in Figure 10 marks early stopping using only 1% of DAPO as validation, with performance on other datasets remaining near-optimal. This provides us with a simple but effective way to realize all the benefits resulting from SRT without the catastrophic performance collapse: namely, one can simply keep a heldout validation dataset aligned with the downstream tasks of interest, and perform early stopping based on performance on this validation dataset.

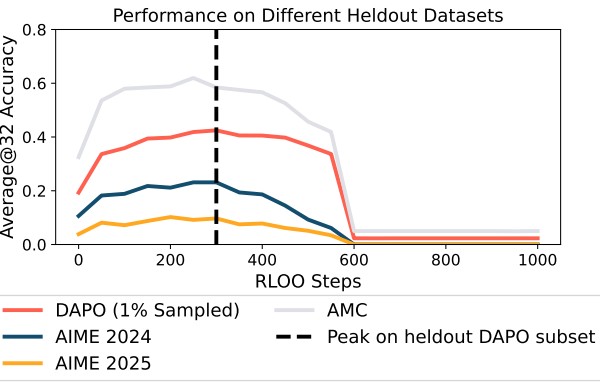

Figure 10: **Early stopping is effective.** The peak performance occurs at nearly the same point for all held-out sets, so using any would be effective for early stopping.

### 5.2 Self-Training with Curriculum Learning

Our third approach, curriculum learning, is motivated by the observation that the model experiences earlier collapse when training on the difficult DAPO dataset compared to the simpler MATH-12K dataset. The intuition is that, on a more challenging dataset, the model finds it easier to abandon its pretrained knowledge in favor of optimizing self-consistency rather than genuinely learning to solve the underlying task.

We leverage this hypothesis to implement a curriculum learning strategy (Bengio et al., 2009; Andrychowicz et al., 2017; Portelas et al., 2020; Florensa et al., 2017; Song et al., 2025b; Lee et al., 2025; Tajwar et al.,

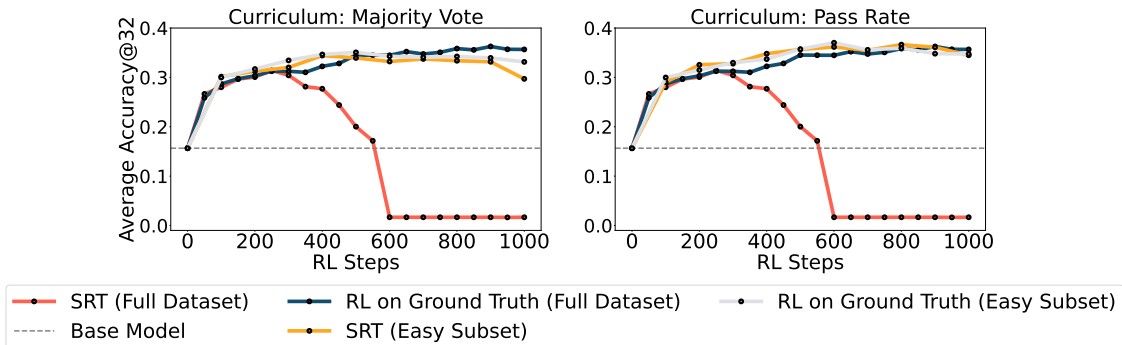

Figure 11: **(Curriculum-Based Self-Training)** Performance of SRT on curated subsets containing the easiest 1/3 of prompts from the DAPO dataset, selected based either on model pass rate or frequency of the majority vote. Training on these easier subsets prevents reward hacking even after extensive training (3 epochs), demonstrating the effectiveness of curriculum learning strategies in sustaining continual model improvement.

2025) by identifying the 'easiest' subset of the DAPO dataset. To be precise, we retain 1/3-rd of the easiest DAPO prompts selected according to two distinct metrics:

(1) **Pass rate of the base model**, which utilizes ground-truth labels.

(2) **Frequency of the majority vote**, which does not require ground-truth labels.

Figure 11 shows that training on these easier subsets significantly delays the onset of reward hacking, allowing for continuous improvement even across multiple epochs. Remarkably, performance on these curriculum subsets reaches levels comparable to standard RL training with ground-truth labels on the entire DAPO dataset. More importantly, **we did not observe training collapse even after 3 epochs of training on 1/3rd of "easy" DAPO** dataset. These promising results suggest that curriculum strategies may further extend the benefits of SRT, which we leave as a future research direction. Additional experiments related to curriculum learning can be found in Appendix J.

## 6 Related Works

**Self Improving LLM.** Previous works (Zelikman et al., 2022; Wang et al., 2023b; Huang et al., 2023; Madaan et al., 2023; Chen et al., 2024; Gulcehre et al., 2023; Singh et al., 2024; Ni et al., 2023; Hwang et al., 2024; Havrilla et al., 2024; Pang et al., 2024a) have demonstrated the feasibility of LLMs' self-improvement over their previous iteration by training on data distilled by the previous instances of the model. Most of these approaches usually have data filtering/reranking step in the pipeline, which is often performed by the model itself (Wu et al., 2025b) or by training another (Hosseini et al., 2024) verifier model. Particularly, (Huang et al., 2023; Wang et al., 2023a; Prasad et al., 2024) demonstrated the feasibility of using majority voting and self-consistency to filter chain-of-thought traces that, when used as SFT training data, improve the LLM performance on downsteaming tasks. A concurrent work, (Zhao et al., 2025a), proposes a self-evolution, self-play pipeline where an LLM generates coding problems of appropriate difficulty, solves and trains on them using RLVR. The model-generated solutions are validated by a code executor in the loop. Previous works have studied self-improvement by generating labels through majority voting (Prasad et al., 2024; Huang et al., 2023), but these works are typically confined to one or a few rounds of SFT or DPO (Rafailov et al., 2024). This essentially involves distilling majority voting labels from a fixed policy into the current policy over each round of training. In contrast, we explore *online RLVR's* potential in self-improving LLMs where the label generating policy evolves after every gradient step. A few concurrent works (Chen et al., 2025; Prabhudesai et al., 2025; Zhao et al., 2025b; Shao et al., 2025) have also explored various forms of self-rewarding mechanisms through majority voting or a similar metric of self-consistency (e.g., token/sequence level entropy) in Online RLVR. Finally, in a recent work, (Song et al., 2025b) formalized a generation verification gap as central to the model's ability to self-improve. Similarly, (Huang et al., 2025) proposed a "sharpening" mechanism as the

key to self-improvement. Our SRT pipeline builds on top of both of these intuitions. We refer the interested reader to Gao et al. (2025) for a survey of other works associated with self-evolving LLM agents.

**Online RLVR and Easy to Hard Generalization.** Online reinforcement learning with verifiable reward (RLVR) (Lambert et al., 2025) has emerged as a new paradigm of LLM post-training especially for enhancing math, coding and reasoning performances (OpenAI et al., 2024; DeepSeek-AI et al., 2025; Team et al., 2025; Lambert et al., 2025). Despite the success of the reasoning models, it is still unclear to what extent they can generalize beyond the difficulty of their training data distribution, a problem termed easy to hard generalization (Sun et al., 2024). (Sun et al., 2024) shows that models can be trained to solve level 4-5 MATH(Hendrycks et al., 2021) problems after training using a process reward model trained on MATH level 1-3 dataset. Another work (Lee et al., 2025) explores this question and finds that transformers are capable of easy-to-hard generalization by utilizing *transcendence* phenomenon (Zhang et al., 2024) in the context of simple addition, string copying, and maze solving using small language models.

**Model Collapse and Reward Hacking.** Model collapse is a well-known phenomenon in training on self-generated training data (Alemohammad et al., 2024; Shumailov et al., 2024b;a; Bertrand et al., 2024; Briesch et al., 2025), and multiple approaches related to data mixing, reliable verification, training using contrastive loss using negative samples and curriculum learning have been proposed (Gerstgrasser et al., 2024; Feng et al., 2025; Briesch et al., 2025; Song et al., 2025b; Gillman et al., 2024; Setlur et al., 2024) to prevent models from collapsing, which previous work on LLM's easy to hard generalization (Lee et al., 2025) also utilize. However, in RL paradigm, we do not directly do supervised fine-tuning on model-generated data, and it remains an open question to what extent the previous findings of model collapse apply to our RLVR setting. In our work, we show that models trained using RL on self-labeled data often suffer from actor collapsing due to reward hacking (Amodei et al., 2016; Denison et al., 2024) and propose a few strategies to mitigate it. Finally, a concurrent and complementary work, Zhang et al. (2025b), has also demonstrated the failure of various self-reward mechanisms to sustain self-improvement under prolonged training, which further validates the observations in this work.

**Data Efficient RLVR.** Several concurrent works look into the data efficiency of the RLVR pipeline. Notably, (Wang et al., 2025) shows that by just training on one example, the model can achieve performance equivalent to training on 1.2k examples from the DeepScaleR dataset (Luo et al., 2025). Similarly, TTRL (Zuo et al., 2025), also proposes a label-free online RLVR paradigm in a test-time setting, similar to SRT. TTRL's test time training to improve performance can be considered an extension of SRT, and we also explore this paradigm in Section I.1. Moreover, our work focuses extensively on how far self-rewarded training can be taken and does an in-depth analysis of the training dynamics in both real-world and synthetic tasks.

## 7 Limitations and Conclusion

In this work, we examine a simple strategy of leveraging an LLM's self-consistency to train it via an RL framework. Our experiments on synthetic reasoning tasks with controllable difficulty levels demonstrate the promise of such a framework: not only can LLMs improve their performance on these tasks, but they can also improve the quality of their self-supervision for the next training step. Our analysis, however, also reveals the limitation of leveraging self-consistency as training reward: prolonged training under such a framework can lead to reward hacking and complete model collapse. Future investigations could explore how to develop more robust forms of verification, extend self-verification to other domains like coding, and curriculum learning strategies that expose the model to problems of only the right difficulty. Additionally, leveraging LLM-as-judges or generative verifiers to improve the training signal, or employing additional consistency regularization between the chain-of-thought and the final answer to mitigate reward hacking can be promising future research. We leave these and other promising directions for sustained self-improvement for future work.

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

# A  Definition of Generation-Verification Gap

The single-generation accuracy is defined as:

$$\text{Acc}_{\text{gen}}(\theta) = \mathbb{E}_{x \sim \mathcal{X}, y \sim \pi_\theta(\cdot|x)}[\mathbf{1}(y = y^*)],$$

where $y^*$ is the correct solution. A verifier function $f$ selects one candidate from multiple generations:

$$f\left(x, \{y^{(1)}, \ldots, y^{(n)}\}\right) \in \{y^{(1)}, \ldots, y^{(n)}\}.$$

We define verification accuracy as:

$$\text{Acc}_{\text{ver}}(\theta, n) = \mathbb{E}_{x \sim \mathcal{X}}\left[\mathbf{1}\left(f\left(x, \{y^{(1)}, \ldots, y^{(n)}\}\right) = y^*\right)\right].$$

We say that a positive *generation-verification gap* occurs whenever $\text{Acc}_{\text{ver}}(\theta, n) > \text{Acc}_{\text{gen}}(\theta)$. Such a gap indicates the verifier's greater proficiency in recognizing correct solutions within a set of candidates compared to the generator independently generating correct answers.

## A.1  Generation Verification Gap Through Majority Voting

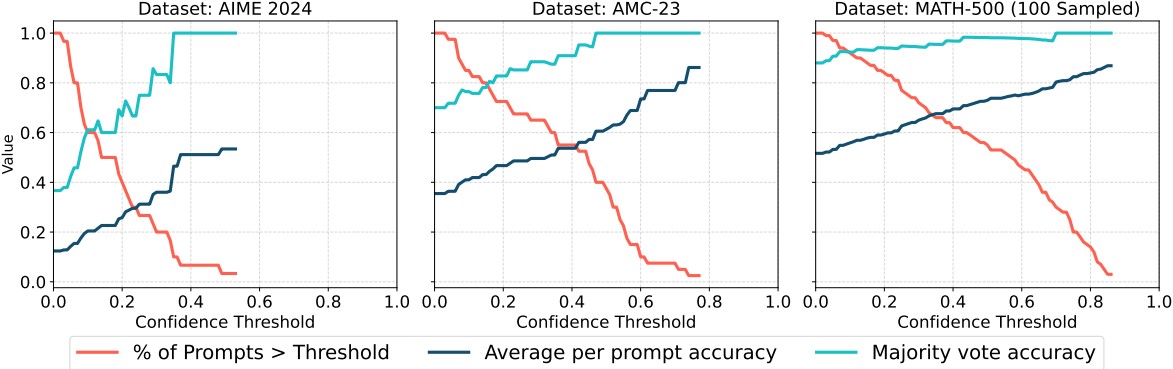

Figure 12: Three of our test datasets display evidence of generation verification gap through majority voting. **The positive gap between majority voting accuracy and per prompt accuracy means LLMs can utilize this gap as a learning signal to improve their average accuracy**. x axis refers to the threshold cut off for self-consistency (proportion of answers that are majority voted answers). Higher x value refers to more self-consistent LLM outputs. For example, at `x = 0.4`, we only keep LLM responses where at least 40% of the (properly parsed) responses were the majority voted answer. On y axis, **average per prompt accuracy** refers to the average number of correct answers out of the (successfully parsed) responses, across 32 generations per question (computed on a per prompt basis and then averaged over the dataset). **Majority vote accuracy** refers to how often the majority vote is the correct answer for that prompt (then averaged over the dataset). **% of Prompts** was computed over all 32 generations (*not* on the parsable answers for a fair comparison). If there is no prompt left over a certain threshold, the line plot ends there.

# B  Details on Implementation and Training

## B.1  RL Algorithms

We use two RL algorithms in this work: RLOO(Ahmadian et al., 2024; Kool et al., 2019) and GRPO (Shao et al., 2024). Recent work (Oertell et al., 2024) has shown that heuristic policy gradient algorithms like GRPO can produce unexpected results by increasing or decreasing reasoning performance even under random rewards, where policy gradient should be zero in expectation, and that RLOO does not have this problem. Since SRT is compatible with both RL algorithms, we experiment with both and observe no noticeable difference in the resulting behavior.

In our implementation (`verl`), we use the following RL objective for both RLOO and GRPO:

$$\mathcal{J}_{\text{GRPO}}(\theta) = \mathbb{E}_{x \sim \mathcal{D}, \ \{y_i\}_{i=1}^{G} \sim \pi_{\theta_{\text{old}}}(\cdot|x)} \left[ \frac{1}{G} \sum_{i=1}^{G} \frac{1}{|y_i|} \sum_{t=1}^{|y_i|} \min \left( w_{i,t}(\theta) \hat{A}_{i,t}, \right. \right.$$
$$\left. \left. \text{clip}(w_{i,t}(\theta), \ 1 - \varepsilon, \ 1 + \varepsilon) \hat{A}_{i,t} \right) \right]$$

where $w_{i,t}(\theta)$ is the importance ratio, defined as:

$$w_{i,t}(\theta) = \frac{\pi_\theta(y_{i,t} \mid x, \ y_{i,<t})}{\pi_{\theta_{\text{old}}}(y_{i,t} \mid x, \ y_{i,<t})}$$

Since we operate fully on-policy, i.e., one RL step per one batch of generated rollouts, this is always one in our experiments. The same advantage defined at a sequence level is applied to each token in the sequence, so henceforth we will drop the $t$ from the notation as well.

The main difference between GRPO and RLOO then stems from their use of different advantage functions. RLOO objective uses the following advantage function:

$$\frac{1}{G} \sum_{i=1}^{G} [R(y_{(i)}, x) - \frac{1}{G-1} \sum_{j \neq k} R(y_{(j)}, x)]$$

whereas GRPO uses the following advantage function:

$$\hat{A}_i = \frac{r(x, y_i) - \text{mean}\left(\{r(x, y_i)\}_{i=1}^{G}\right)}{\text{std}\left(\{r(x, y_i)\}_{i=1}^{G}\right)}$$

Here $G$ is the number of online samples generated. Both RLOO and GRPO creates a dynamic baseline for each sample without needing a separate value function, effectively estimating the expected return on-the-fly during training. Not having a value networks makes the training much simpler for both algorithms.

In our implementation, we did not add KL penalty to the loss function, rather to the reward itself while running RLOO, following recent work such as Tang & Munos (2025). In verl framework, this can be configured using `algorithm.use_kl_in_reward=True` and `actor_rollout_ref.actor.use_kl_loss=False`. However, this does not work for GRPO due to advantage normalization by the standard deviation, and so for GRPO we add KL penalty to the loss function directly. To estimate KL penalty, we use the low variance KL estimator proposed by Schulman (2020):

$$\mathbb{D}_{\text{KL}}(\pi_\theta || \pi_{\text{ref}}) \approx \frac{\pi_{\text{ref}}(y|x)}{\pi_\theta(y|x)} - 1 - \log\left(\frac{\pi_{\text{ref}}(y|x)}{\pi_\theta(y|x)}\right)$$

**Sampling.** For all experiments, we kept the generation `temperature` to 1.0, `top_k` to -1, and `top_p` to 1 for rollouts generated during RL rollouts. Decoding temperature used for validation varies in different settings, see Appendix B.3 and D for more discussion. We cut off maximum prompt length at 1024 and maximum response length to 3072 (note: Qwen2.5-Math-7B models support a maximum context window of 4096).

## B.2 GPU Infrastructure

All experiments in this work were conducted using either a single node consisting of 8 NVIDIA H200 GPUs (141 GB of GPU memory per GPU) or a single node consisting of 4 NVIDIA GH200 GPUs (96 GB of GPU memory per GPU). All experiments can be replicated in single-node training, and we did not, in fact, utilize multinode training. In total, this work consumed ~15000 GPU hours (including preliminary studies and failed runs). All the final results listed in this paper can be replicated within 2000 H200 GPU hours.

## B.3 Details on Training Settings

We choose Family Relationships, Bitwise Arithmetic, and Knights & Knaves (Xie et al., 2024) tasks from Reasoning Gym (Stojanovski et al., 2025) for our experiments. Examples for each task is shown in Appendix L.

For the **Bitwise Arithmetic** task, *Level* refers to the `difficulty` parameter. The model was first trained with level 2 data for 950 steps, reaching 97% accuracy. We then used this initialized model to train with SRT on levels 3 and 4.

For the **Family Relationships** task, we trained a model to 99% accuracy on the level 4 dataset. Here, *Level* corresponds to the parameters `min_family_size` and `max_family_size`, both set to 4. We then applied SRT on level 5.

For the **Knights & Knaves** task, we varied only the `n_people` parameter as the difficulty control. We first trained a model with difficulty level 2 to 99% accuracy, and then used that checkpoint to further experiment with SRT, climbing from level 2 to 3, 3 to 5, 5 to 7 and 7 to 9. **We only report levels 7 and 9 in the paper since they are the highest level difficulty among our experiments**.

Across all multi-level experiments, we applied SRT progressively. For example, in Bitwise Arithmetic we trained on level 2 with ground truth supervision; then, starting from the level 2 checkpoint, we applied SRT on level 3; finally, we repeated SRT again on level 4 using the checkpoint from level 3. For comparing against **SRT with fixed teacher**, we use the same starting policy (Qwen3-4B-Base trained with ground-truth on the easiest difficulty level on each task) and generate the same number of rollouts per prompt using temperature 1.0 and perform majority voting among these rollouts to generate our pseudo-labels.

**Default training hyperparameters for Reasoning Gym tasks.** For all Reasoning Gym experiments, we used the `Qwen3-4B-Base` model, with GRPO as the main algorithm. The learning rate was set to `1e-6` and the KL penalty to `0.0001`. For all experiments, we used 32 rollouts per prompt for training and 16 rollouts for evaluation.

**Default training hyperparameters for Math Datasets.**

- **Qwen2.5-Math-7B**: Learning rate $10^{-6}$, KL penalty coefficient 0.001, decoding temperature for training and evaluation rollouts 1.0, top-p 1.0, and no top-k sampling.

- **Qwen3-14B-Base**: Same default hyperparameter setting as Qwen2.5-Math-7B.

- **Llama-3.1-8B-Instruct**: Learning rate $10^{-7}$, KL penalty coefficient 0.001, decoding temperature for training is 1.0 and evaluation is 0.0, top-p 1.0, and no top-k sampling. We subsample the Big-Math-RL-Verified dataset to only keep prompts that has average Llama-3.1-8B-Instruct pass rate between 0.3 and 0.7, since the model is unable to improve during training otherwise.

- **Deepseek-Math-7B-Instruct**: Learning rate $10^{-7}$, KL penalty coefficient 0.001, decoding temperature for training is 1.0 and evaluation is 0.7 (following official protocol), top-p 1.0, and no top-k sampling.

# C   Additional Experimental Results

## C.1   Qwen2.5-Math-7B

Here we compare the performance of training Qwen2.5-Math-7B with SRT and RL with ground truth on each individual test set for the sake of completeness. Figures 13, 14, and 15 show the detailed results when we train on DAPO, MATH-12K, and AIME (1983-2023) respectively. Additionally, when training on MATH-12K and DAPO, we also evaluate the intermediate checkpoints on the heldout set MATH-500, which is reported in Figure 14. **Since MATH-500 contains 500 examples, calculating average@32 accuracy becomes expensive, and hence we could not use it as a test set for all our training setups**.

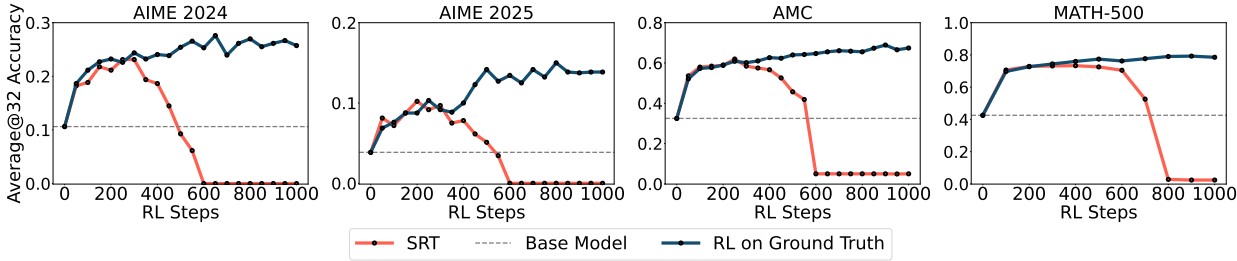

Figure 13: **(Individual test set performance during training on DAPO)** We record the average@32 accuracy during training Qwen2.5-Math-7B on DAPO, on three heldout test sets: AIME 2024, AIME 2025 and AMC. In all three cases, SRT performance collapses, while training with ground truth keeps improving steadily.

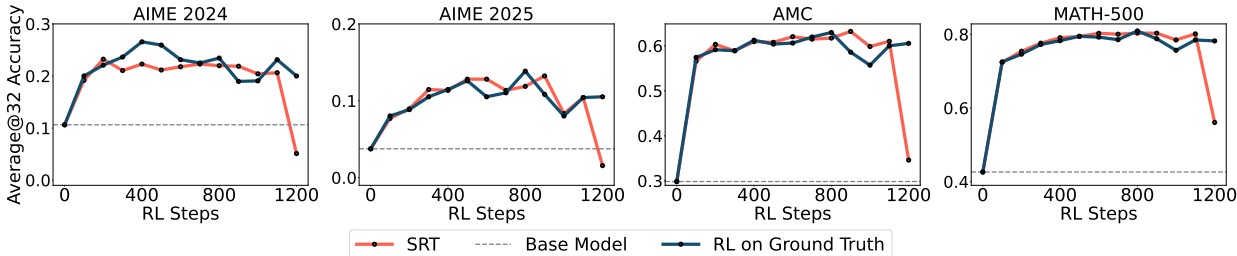

Figure 14: **(Individual test set performance during training on MATH-12K)** We record the average@32 accuracy during training Qwen2.5-Math-7B on MATH-12K, on three heldout test sets: AIME 2024, AIME 2025 and AMC. We also evaluate intermediate checkpoints on MATH-500 since we are training on MATH-12K (we could not do this for other training datasets due to a lack of computational resources). In all 4 heldout test sets, SRT results in similar performance gain as one would obtain from training with ground truth labels. However, performance collapses after 1200 RL steps, similar to our other observations.

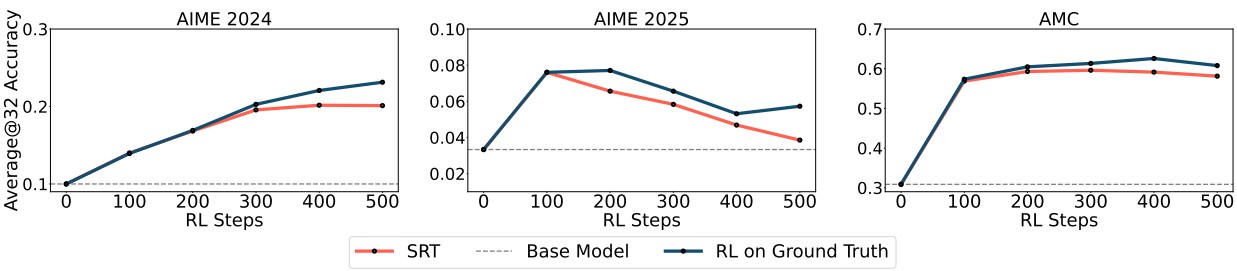

Figure 15: **(Individual test set performance during training on AIME (1983-2023))** We record the average@32 accuracy during training Qwen2.5-Math-7B on AIME (1983-2023), on three heldout test sets: AIME 2024, AIME 2025 and AMC. SRT performs similarly or better compared to training with ground truth labels over 10 epochs of training.

## C.2 Qwen3-14B-Base

In addition to Qwen2.5-Math-7B, we apply our algorithm on another LLM — namely Qwen3-14B-Base (Yang et al., 2025). We choose the base model since it has not gone through additional post-training on reasoning tasks, unlike the Qwen3-14B model. Additionally, this is a significantly larger model with a different pre-training, making it suitable for testing our algorithm's effectiveness.

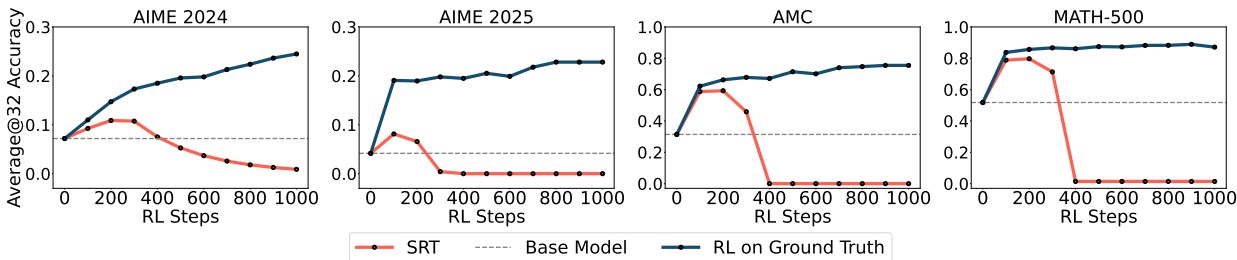

Figure 16: **(Individual test set performance during Qwen3-14B-Base on DAPO)** We record the average@32 accuracy during training a Qwen3-14B-Base model on DAPO, on four heldout test sets: AIME 2024, AIME 2025, AMC, and MATH-500.

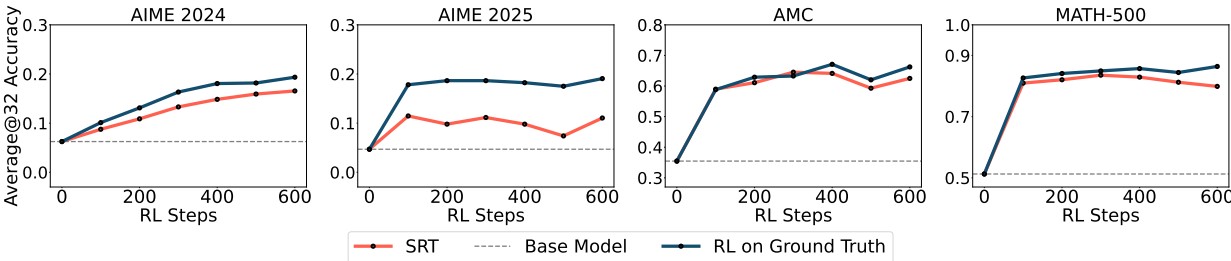

Figure 17: **(Individual test set performance during training Qwen3-14B-Base on MATH-12K)** We record the average@32 accuracy during training a Qwen3-14B-Base model on MATH-12K, on four heldout test sets: AIME 2024, AIME 2025, AMC, and MATH-500.

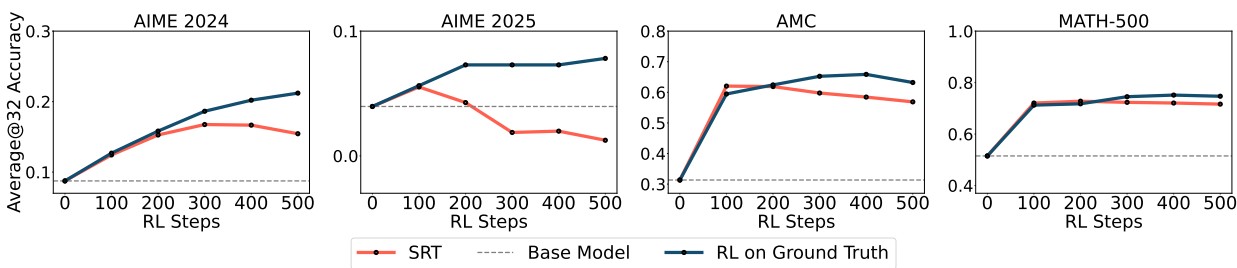

Figure 18: **(Individual test set performance during training Qwen3-14B-Base Model on AIME (1983-2023)** We record the average@32 accuracy during training a Qwen3-14B-Base model on AIME (1983-2023), on four heldout test sets: AIME 2024, AIME 2025, AMC, and MATH-500.

Figures 16, 17, and 18 shows our results with DAPO, MATH-12K, and AIME (1983-2023) used as training dataset respectively. Our experiments with Qwen3-14B-Base mostly follows similar patterns as Qwen2.5-Math-7B: SRT maintains stable performance on MATH-12K, mixed results on AIME (1983-2023), and performance collapse on DAPO.

# D  Effect of Decoding Temperature (Qwen2.5-Math-7B)

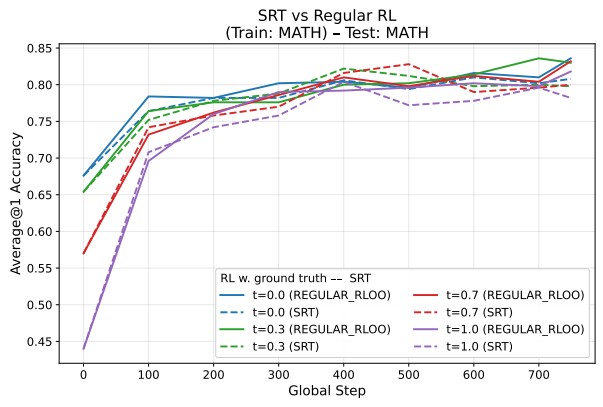
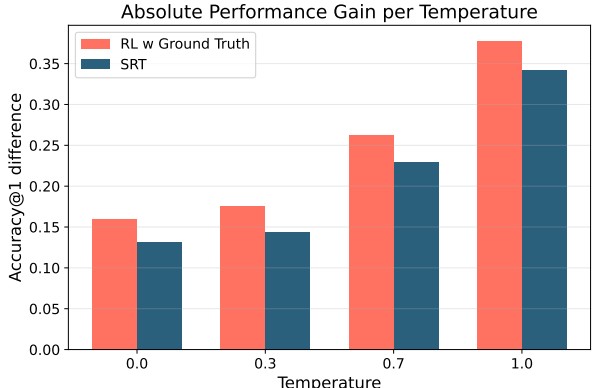

(a) Math-500 evaluation accuracy during training, when decoded under various temperature (Qwen2.5-Math-7B).

(b) Absolute performance improvement after one epoch of training when decoded under various temperatures (Qwen2.5-Math-7B).

Figure 19: Our method (SRT) performs consistently regardless of decoding temperature for validation (Figure 19a). All experiments are run using Qwen2.5-Math-7B as the base model, trained on MATH-12K and tested on MATH-500 measured in terms of average accuracy@1. Notice that even though the performance is low initially at high temperature, at the later stages, they plateau around the same point. Figure 19b shows the absolute gain when decoded under different temperatures. Note that decoding with higher temperature might give the impression of a larger gain compared to low-temperature decoding. However, the evaluation curves (Figure 19a) during training resulting from SRT and RL with ground truth look almost identical regardless of decoding temperature.

# E  Additional Self-Training Metrics

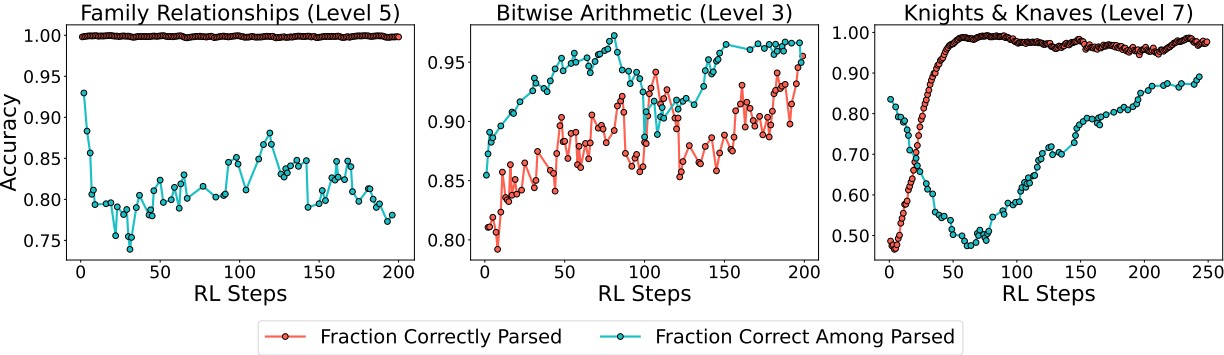

Figure 20: **(Tracking format following success rate during SRT-training on Reasoning Gym)** In order to track whether SRT is teaching reasoning strategies beyond just formatting the final answer correctly, we track two additional metrics throughout SRT-training: fraction of generations among all generations where the final answer is parseable and fraction of generations among those that are parseable where the final answer is correct. We see that due to training with RLVR on an easier level of difficulty, the starting policy can already format most generations correctly, and in the case of Knights & Knaves, fraction of correct responses keeps increasing even after fraction of properly formatted (and thus parseable) responses have saturated. This shows that the model learns reasoning strategies beyond formatting rules.

We are interested to know if SRT teaches actual reasoning strategies beyond just proper formatting rules necessary for extracting the final answer. To do so, we track two additional metrics throughout SRT-training: fraction of generations among all generations where the final answer is parseable and fraction of generations among those that are parseable where the final answer is correct. Figure 20 summarizes our findings on

Reasoning Gym: We see that due to training with RLVR on an easier level of difficulty, the starting policy can already format most generations correctly, and in the case of Knights & Knaves, fraction of correct responses keeps increasing even after fraction of properly formatted (and thus parseable) responses have saturated.

# F   Can SRT be Applied in Non Math Domain?

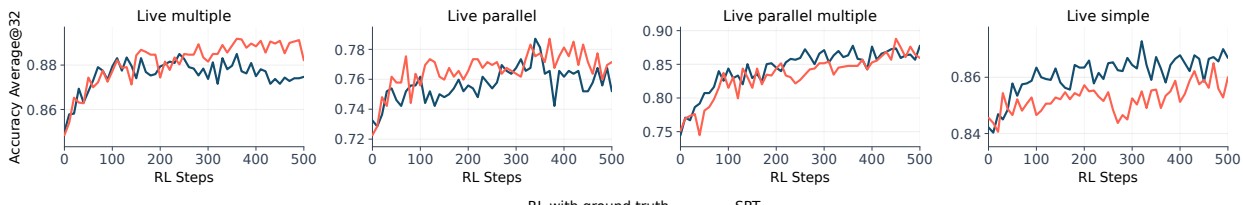

Figure 21: **(SRT Makes the Model Better at Tool Calling)** Here, for the four held-out evaluation datasets of the Berkeley Function Calling Leaderboard V2, SRT performs as well as RL with ground truth. Here, the model under training is Qwen3-4B, which was already trained for tool calling; thus, the initial performance is high. We used the exact same tool, calling formatting required for this model to format our dataset and chat template.

Most of the experiments in our work has centered around math and synthetic reasoning puzzle tasks. One core question remains whether SRT can be applied to non math domain where majority voting is still well defined. We found tool calling to be one of the domains.

**Majority-vote reward.**   Similar to math majority voting, for each training prompt, we parse every valid rollout into a canonical list of tool calls and select the most frequent canonical output as the self-consistency label. Canonicalization ignores JSON whitespace and object-key order and, for parallel categories, also ignores the order of tool calls; unparseable outputs do not participate in the vote.

**Training data.**   We use the single-turn AST portion of the Berkeley Function Calling Leaderboard (BFCL) (Yan et al., 2024). More precisely, the training set uses the Python categories introduced in BFCL V1, while the evaluation set uses categories from BFCL V2 Live.

We train on 1,000 single-turn examples from BFCL V1. The category composition is:

| BFCL category | Number of examples |
| --- | --- |
| simple_python | 400 |
| multiple | 200 |
| parallel | 200 |
| parallel_multiple | 200 |
| Total | 1,000 |

The terminology is:

- **parallel**: multiple calls to the same supplied function.

- **multiple**: choose the correct function from multiple candidates.

- **parallel_multiple**: select multiple functions and make multiple calls in one response.

Each prompt contains a system message with the available function definitions and a user message containing the request.

**Evaluation data.**   We evaluate on 240 gold-labeled examples drawn from four BFCL Live categories:

| BFCL Live category | Used | Full set | Selection |
| --- | --- | --- | --- |
| live_simple | 100 | 258 | First 100 |
| live_multiple | 100 | 1,053 | First 100 |
| live_parallel | 16 | 16 | Full category |
| live_parallel_multiple | 24 | 24 | Full category |
| Total | 240 | 1,351 | |

Live categories such as live_simple and live_multiple come from BFCL V2 Live. They were contributed from real enterprise and open-source use cases, with newer and less synthetic function specifications and user queries. We subsampled first 100 data points from the first two categories to make the evaluation faster since our evaluation is averaged over 32 inferences.

**Model.** We used the Qwen3-4B model with the default thinking mode on for our experiments. The full system prompt for this experiment can be found in the Appendix M.

**Results.** As shown in Figure 21, label free SRT performs almost as well as RL with ground truth for all four held out evaluation datasets. Here we trained the Qwen3-4B model, which already has some tool calling capability and thus the initial performance starts quite high. We trained with a batch size of 16, with the number of generations per prompt = 32. The evaluation uses 32 rollouts sampled at temperature 1.0.

# G  More Details on Baselines

**Baseline Implementation.**    For all three methods (SFT, DPO (Rafailov et al., 2024), ScPO (Prasad et al., 2024)), we sweep over three learning rates ($10^{-5}$, $10^{-6}$ and $10^{-7}$) and pick the checkpoint with the highest validation score. The best checkpoint with highest validation in the SFT stage has been used to initialize the DPO/ScPO training. We also train DPO with the above mentioned learning rates and picked the best score. For DPO and ScPO (Prasad et al., 2024), we used $\beta = 0.1$, which we also found through sweep over (0.1, 0.3 and 0.5). Moreover, we add a negative log-likelihood loss with weight 1.0 to the DPO and ScPO losses to stabilize them, similar to RPO (Pang et al., 2024b). We do not train for more than 1 epoch to prevent overfitting/unintentional unalignment (Tajwar et al., 2024; Razin et al., 2025) and fair comparison with SRT.

| Train Dataset | Method | AMC/AIME | MATH500 |
|---|---|:---:|:---:|
| **MATH** | SFT | 0.18 | 0.75 |
| | ScPO | 0.20 | 0.72 |
| | DPO | 0.23 | 0.74 |
| | **SRT (Ours)** | **0.32** | **0.80** |
| **DAPO** | SFT | 0.18 | 0.75 |
| | ScPO | 0.20 | 0.72 |
| | DPO | 0.21 | 0.76 |
| | **SRT (Ours)** | **0.31** | 0.75 |
| **Base Model** | Accuracy | 0.15 | 0.42 |
| | Majority@32 Acc | 0.20 | 0.79 |

Table 2: Comparison of different methods trained on either the MATH or DAPO dataset. Performance is evaluated using two metrics: (1) the average accuracy over 32 generations per sample across AMC 2023, AIME 2024, and AIME 2025, collectively denoted as AMC/AIME, and (2) average accuracy@1 computed over MATH500. Notice that the majority@32 accuracy scores are not directly comparable with the other accuracy metrics listed in the table.

**Dataset Curation**   For DPO and ScPO we labeled the most consistent response as the positive example and the least consistent response as the negative example for each question. Moreover, we only kept the instances where $w(x)$, the variance based weighing parameter (Prasad et al., 2024), was greater than 2.

# H    Detailed Experiment Results on Different Training Settings

## H.1    GRPO vs RLOO

Figure 8 shows our experiment comparing how SRT behaves with different RL algorithms. In particular, we test two algorithms: GRPO and RLOO. While GRPO seems to achieve higher performance, both GRPO and RLOO training with SRT-reward leads to model collapse at similar number of steps — showing that the choice of the RL algo does not influence model collapse.

## H.2    Different KL Penalty Coefficients

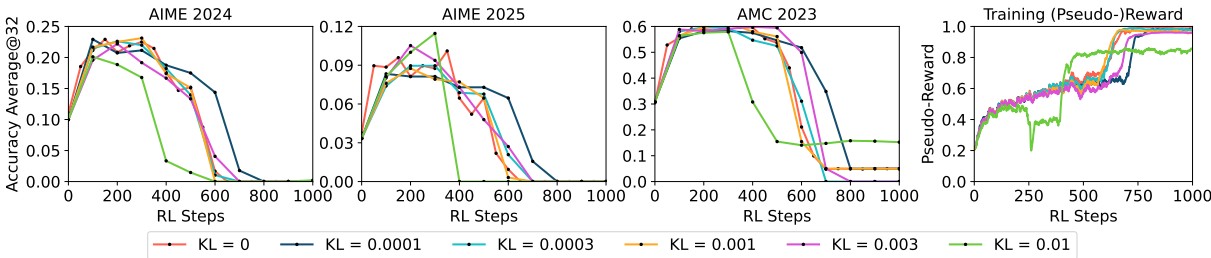

Figure 22: **(SRT with different KL penalty coefficients)** We compare the behavior of SRT with different KL penalty coefficients and report performance on three test datasets. The rightmost plot plots the training reward for all these training runs. All experiments here use a Qwen2.5-Math-7B model trained with RLOO on the DAPO dataset, with the hyperparameters other than the KL penalty coefficient being the default ones described in Appendix B.3. We find that applying a KL penalty does not prevent model collapse; rather, it seems to accelerate it.

The most straightforward way of preventing reward hacking is to add a strong KL penalty to the training objective. In Figure 22, we explore this idea: to our surprise, we don't find a higher KL penalty coefficient to delay or prevent model collapse. We attribute this to the reward hacking training signal being too strong to be overcome by the KL regularization.

## H.3 Learning Rate

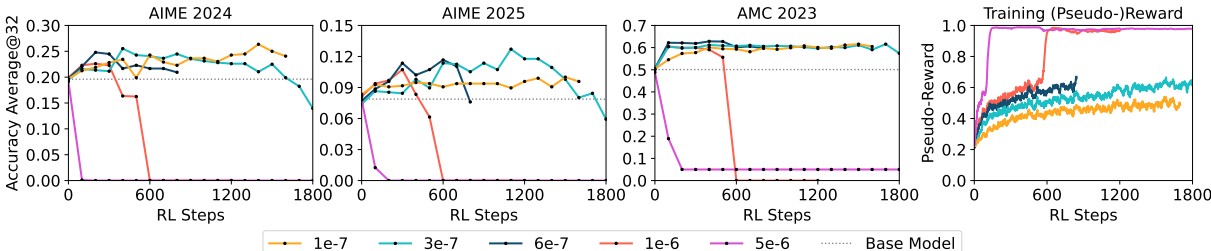

Figure 23: **(SRT with different learning rate)** We compare the behavior of SRT with different learning rate and report performance on three test datasets. The right most plot shows the training reward as the training progresses. All experiments here use a Qwen2.5-Math-7B model trained with RLOO on the DAPO dataset, with the hyperparameters other than learning rate being the default ones described in Appendix B.3. Lower learning rate tends to delay model collapse in our experiments, although the peak performance remains the same. Notice that a higher learning rate ($5e^{-6}$) hastens the model collapse even further and leads to almost immediate model collapse. All the other experiments in our work uses $5e^{-6}$ learning rate unless specified otherwise.

Another common hyperparameter to tune is the learning rate. To investigate the effect learning rate has on SRT, we finetune a Qwen2.5-Math-7B model with different learning rates using SRT on the DAPO dataset, with all other hyperparameters kept fixed at their default values described in Appendix B.3. Figure 23 shows our empirical findings: lowering learning rate seems to prevent model collapse within our training budget. However, we notice performance degradation on the harder AIME 2024 and 2025 datasets within our training budget, and hypothesize that prolonged training with SRT, even with a considerably lower learning rate, would still lead to model collapse. We could not study this in detail due to computational constraints, and leave this for future work.

## H.4 Different Number of generations per prompt

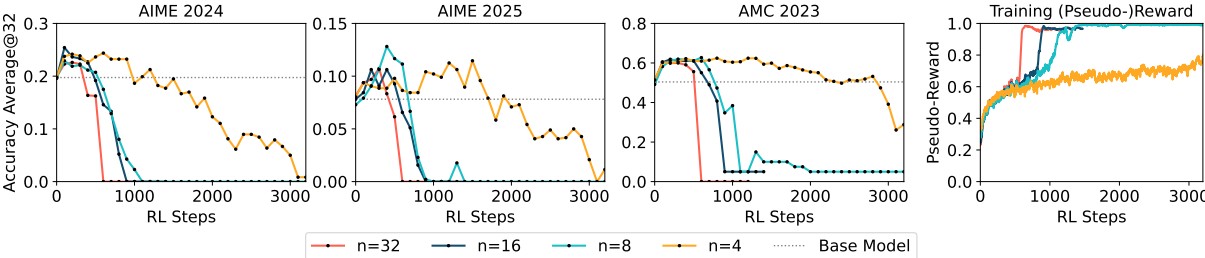

Figure 24: **(SRT with number of generations per prompt)** We compare the behavior of SRT with different number of generations/rollouts per prompt and report performance on all test datasets. All experiments here use a Qwen2.5-Math-7B model trained with RLOO on the DAPO dataset, with the hyperparameters other than number of generations per prompt kept fixed at the default ones described in Appendix B.3. Surprisingly, lowering the number of generations per prompt seems to delay model collapse; however, the peak performance reached by SRT remains the same.

We observe the most surprising result among our hyperparameter tuning experiments when we vary the number of rollouts per prompt during training. Similarly as before, we finetune a Qwen2.5-Math-7B model with SRT on the DAPO dataset, with all hyperparameters (except number of rollouts per prompt) kept fixed at the their default values described in Appendix B.3. Figure 24 records our results: model collapse happens progressively later in the training run as we lower the number of rollouts per prompt. We hypothesize that generating fewer rollouts makes estimating the "true" majority voting label for a single prompt more noisy. This noisy estimation then makes hacking the reward signal harder as well, thereby delaying model collapse. We have not been able to study this phenomenon in more detail due to computational constraints, and leave studying this for future work.

## H.5 Entropy Coefficient

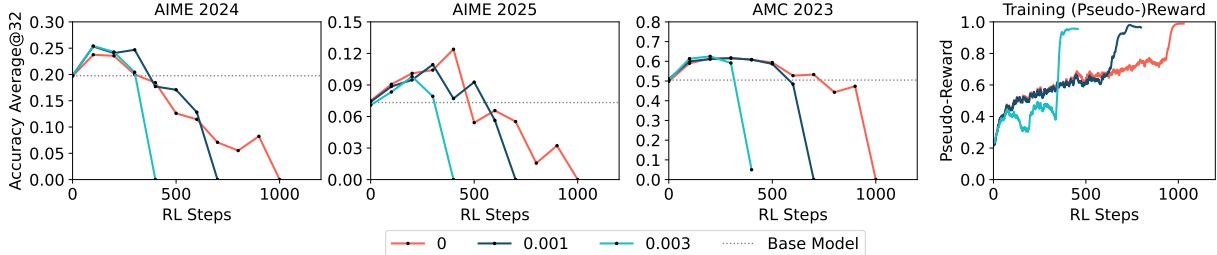

Figure 25: **(SRT with different entropy coefficient)** We compare the behavior of SRT with different entropy coefficients and report performance on all test datasets. All experiments here use a Qwen2.5-Math-7B model trained with RLOO on the DAPO dataset, with the hyperparameters other than entropy coefficient kept fixed at the default ones described in Appendix B.3. Surprisingly, increasing the entropy coefficient hastens model collapse.

The final hyperparameter with which we experiment is adding an entropy loss to our regular RL objective. The modified RL objective is listed below:

$$\mathcal{L}_{\text{entropy-augmented}}(\pi_\theta) = \mathcal{L}_{\text{RL}}(\pi_\theta) - \alpha\mathcal{H}(\pi_\theta) \tag{7}$$

where $\mathcal{L}_{\text{RL}}(\pi_\theta)$ is the regular RL loss objective, $\alpha$ is the entropy coefficient, and $\mathcal{H}(\pi_\theta)$ is the per-token entropy averaged across all tokens in all rollouts. A reasonable hypothesis is that adding entropy to the loss objective will prevent model collapse (Cheng et al., 2025) by discouraging the model to converge to one solution for every prompt. However, Figure 25 shows results in the contrary: adding an entropy term to the loss function and increasing the corresponding coefficient $\alpha$ accelerates model collapse. Upon inspection of the rollouts, we see that increasing $\alpha$ incentivizes the model to generate more random tokens in the rollouts, which maximizes entropy, followed by the same template final answer, which maximizes training (pseudo-)reward. This suggests that a better mechanism to prevent model diversity collapse is needed (Song et al., 2025a; Zhou et al., 2025), which we leave to future work to study.

## H.6 Effects of Data Shuffling

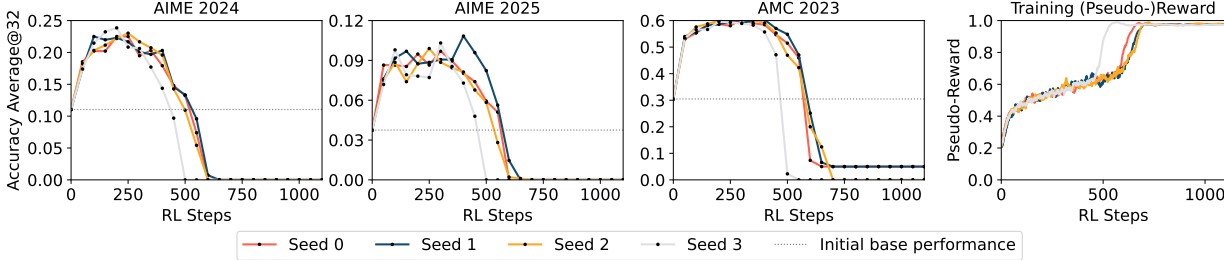

Figure 26: **(SRT with different data shuffling seeds)** Across different data shuffling seeds, SRT performance remains very similar. All experiments here used a Qwen 2.5-Math-7B model trained with RLOOO on the DAPO training dataset.

# I  Test-Time Self-Improvement

## I.1  SRT can be used for test-time training

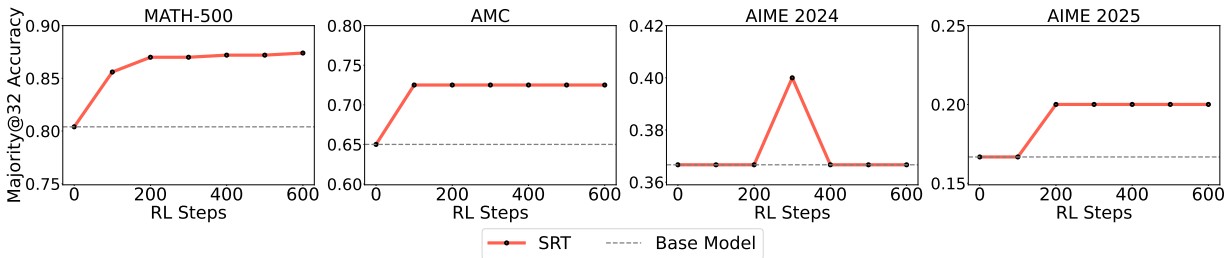

Figure 27: **(Test-Time Self-Training Performance)** Given the test dataset $\mathcal{D}_{\text{test}}$, one can perform SRT on $\mathcal{D}_{\text{test}}$ before making predictions. Our results show that this improves the majority voting performance on $\mathcal{D}_{\text{test}}$ without access to ground truth labels. Notice that the y axis is showing majority@32 accuracy instead of average@32 accuracy, for a fairer comparison with the baseline.

An appealing application of self-training is improving model accuracy via test-time training (Sun et al., 2020; Wang et al., 2021), a direction also explored by the concurrent work of Zuo et al. (2025). Test-time training refers to the procedure of further adapting or fine-tuning a pre-trained model on the actual test set itself, typically without access to labels or ground truth annotations. Applying SRT as a test-time training technique is remarkably straightforward: the unlabeled test set is treated precisely as if it were a training dataset, and SRT is directly applied.

We compare the test-time performance of majority voting after SRT test-time training as well as without any test-time training. Empirically, we observe (Fig 27) that test-time training via SRT provides relatively limited, yet noticeable, performance gains when measured under the maj@32 metric, compared to the popular majority voting baseline applied directly to outputs generated by the base model.

## I.2  Why Doesn't the Performance Collapse during Test-Time-Training?

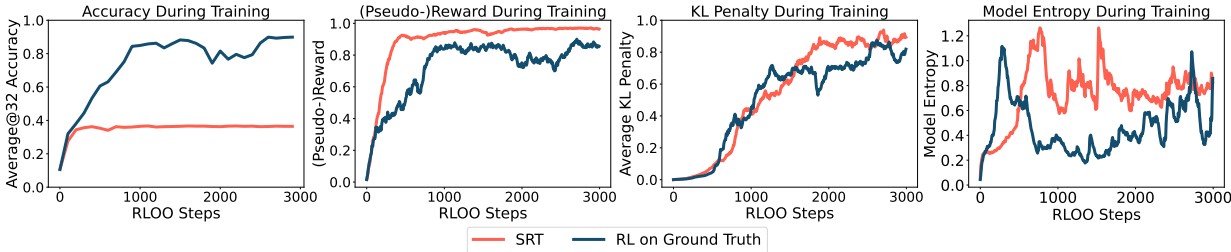

Figure 28: **(Test-Time Self-Training Dynamics)** We apply test-time training on AIME 2024 and observe no performance collapse. However, SRT's performance quickly saturates (leftmost plot), and the pseudo-reward value (second plot) also approaches saturation.

Interestingly, upon completion of test-time training, a visual inspection of model's outputs reveals that the model's predictions still degenerate to a single response for nearly every test prompt—precisely the behavior identified as optimal solution to the SRT objective; however, the test-time accuracy remains high.

We conjecture that test-time self-training is inherently more stable due to crucial differences in dataset size. For example, consider the AIME24 test dataset, which contains only 30 samples for self-improvement. With such a limited sample size, the model quickly converges to a stable majority vote answer on these examples by reinforcing the particular chain-of-thought reasoning that leads to such solutions. After reaching this convergence, SRT ceases to receive meaningful gradient signals for further parameter updates, naturally stabilizing test-time performance (see Figure 28 for test-time training dynamics).

In contrast, during regular training on large-scale datasets, the iterative supply of many fresh samples continually pushes the model to optimize heavily for consistency. In such conditions, the model is incentivized

to adopt an overly simplistic generalization strategy (producing same \boxed{} answer)—eventually collapsing by producing a uniform, prompt-independent prediction.

## J  Additional Experiments with Curriculum Learning

### J.1  Training Dynamics of SRT (Qwen2.5-Math-7B) on the Easy DAPO Subset

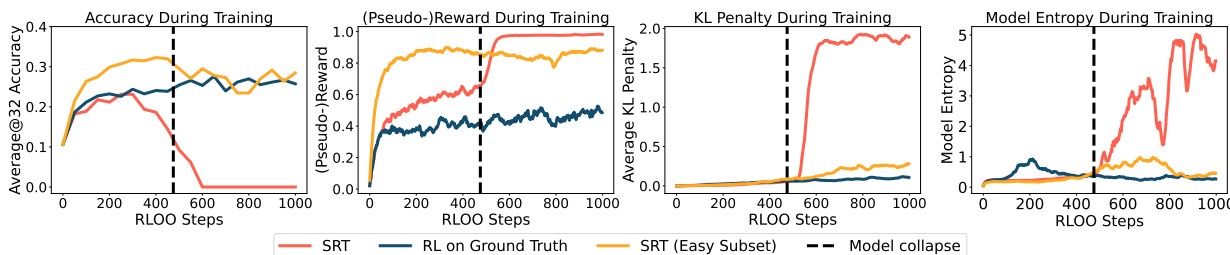

Figure 29: **(Training Dynamics of SRT on the Easy Subset of DAPO, using Qwen2.5-Math-7B)** We show the training dynamics of SRT on the easiest 1/3-rd of the DAPO dataset, chosen by ground truth pass rate of the base model. Compared to SRT on the entire DAPO dataset, SRT on the easier subset does not show any signs of reward hacking, even after taking 3 full passes over the training set.

Figure 7 showed the common signs of reward hacking during SRT-training: namely, sudden drop in accuracy on a held-out dataset, sudden increase in KL penalty, etc. However, we found a simple yet effective way of mitigating reward hacking — simply train on the easiest subset of the training data seems to retain the performance improvement obtained by training on the entire dataset, while preventing reward hacking within the same compute budget. Here we attempt to analyze this phenomenon further, from the lense of the same metrics we recorded in Figure 7.

Figure 29 shows our results on Qwen2.5-Math-7B: SRT-training on the easiest subset does not show the same behavior as training on the full dataset: accuracy on the heldout set does not drop, and KL penalty, while being slightly higher than that of training with ground truth, is still significantly lower than SRT-training on the full dataset. We also see that model entropy does not explode, so the model keeps outputting reasonable responses instead of the degenerate ones resulting from full dataset training. The most intriguing observation is that regarding pseudo-reward (Figure 29, second from left): it very quickly gets very close to 1 and stabilizes around 0.9. This tends to suggest the model gets very little learning signal as the mean of the pseudo reward is already approximately 1, which is probably the reason it does not learn to reward hack within the same compute budget. We leave investigating this further for future work.

### J.2  Training Qwen3-14B-Base on the Easy DAPO Subset

One of our most interesting observations is that simply using the easiest 1/3-rd of the DAPO dataset eliminates the performance collapse within our training budget (it can still happen if one trains more, though we do not observe it). We want to test whether this is still true for a different base model. To do so, we take the same easiest subset used in Figure 29 (so the subset is determined using either the ground truth pass rate or the frequency of the majority answer of a Qwen2.5-Math-7B model) and train a Qwen3-14B-Base model with SRT on this subset. Figure 30 shows the result of our experiments: similar to Qwen2.5-Math-7B model, the Qwen3-14B-Base model also does not exhibit performance collapse within the same training budget.

### J.3  Generating Easy DAPO Subset using Qwen2.5-Math-1.5B

Next, we want to test if the easy subset generation process of our curriculum algorithm itself is reproducible using different base models. To do so, we generate the easy 1/3-rd subset using a Qwen2.5-Math-1.5B model by both majority voting frequency and pass rate. This is in contrast with our earlier sections, especially Figure 11 and Figure 30, where we used a Qwen2.5-Math-7B model for the subset generation. We also train the Qwen2.5-Math-1.5B model on the resulting easy subsets. Figure 31 shows our results — we see the

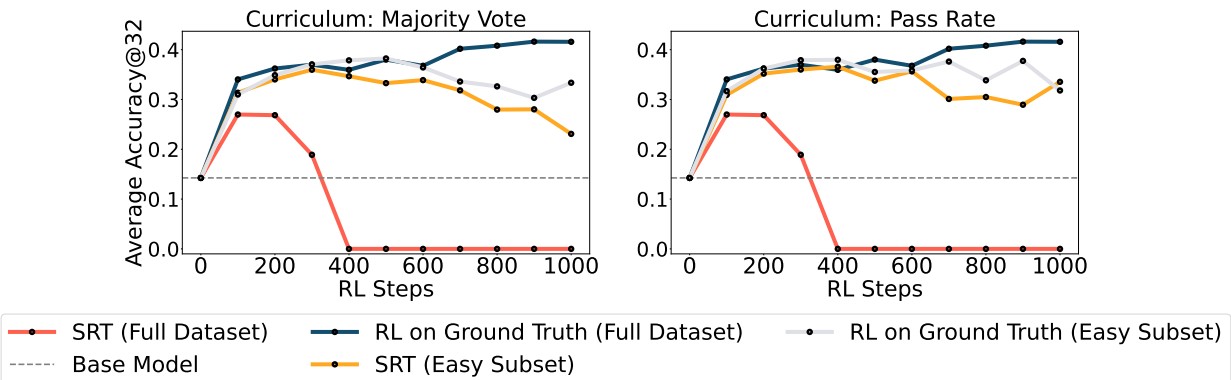

Figure 30: **(Qwen3-14B-Base Trained on the Easy DAPO Subset)** We take the same easy subsets of DAPO used in Figure 29 and train a Qwen3-14B-Base model with SRT on it. We see the same behavior as Qwen2.5-Math-7B (Figure 11), that SRT on the easy subset does not exhibit performance collapse within the same compute budget.

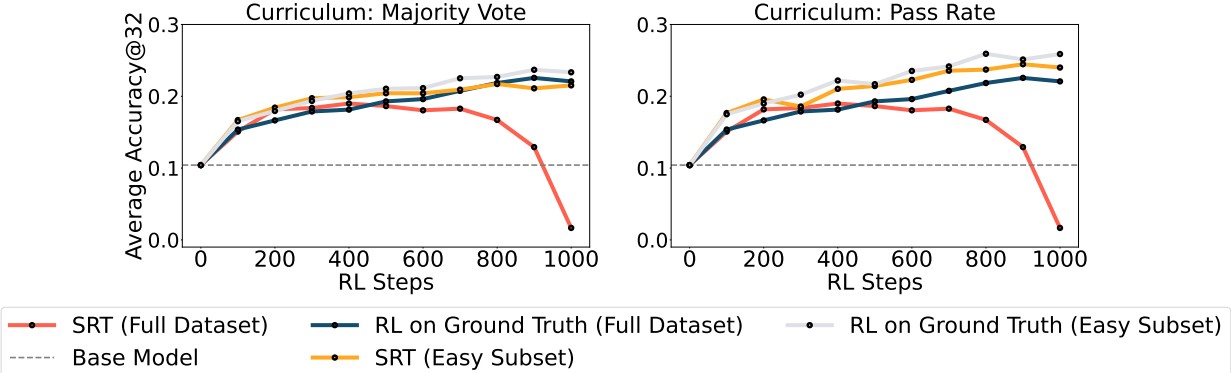

Figure 31: **(Generating Easy DAPO Subset Using Qwen2.5-Math-1.5B)** To see if the curriculum generation process is reproducible, we generate the easy 1/3-rd subset (by both majority voting frequency and pass rate) using a Qwen2.5-Math-1.5B model. This is in contrast with Figure 11 and Figure 30, where we used a Qwen2.5-Math-7B model for the easy subset generation. Furthermore, we train the Qwen2.5-Math-1.5B on the easy subset as before, and make similar observations: training with SRT on a easy subset, even for 3 epochs, does not lead to performance collapse.

same trend as our earlier result, i.e., training on the easy subsets, even for 3 epochs, does not lead to any performance collapse. Moreover, surprisingly, up to our training budget, SRT on the easy subset matches the performance of RL training with ground truth labels.

## K   Detailed Experiment Results using non-Qwen models

To validate the efficacy of SRT on LLMs with different pre-training/post-training routine, we run additional experiments on two more models: Deepseek-Math-7B-Instruct (Shao et al., 2024) and Llama-3.1-8B-Instruct (Grattafiori et al., 2024). Our results are described below.

### K.1   Deepseek-Math-7B-Instruct

We train Deepseek-Math-7B-Instruct on the MATH-12K dataset, and test on AIME 24, AIME 2025, AMC and MATH-500. For training, we use the same hyperparameters use used to train Qwen2.5-Math-7B-Instruct due to lack of compute for running a sweep over possible hyperparameters. We note that we did not find the recommended temperature or other sampling parameters for the Instruct model in (Shao et al., 2024), but their base models were evaluated with temperature 0.7, so we choose temperature 0.7, top-p 1.0 and no top-k sampling for our evaluations. Figure 32 shows our results: we see similar trends as our earlier experiments, where SRT initially matches performance gain from RL with ground truth, but then leads to performance collapse after prolonged training.

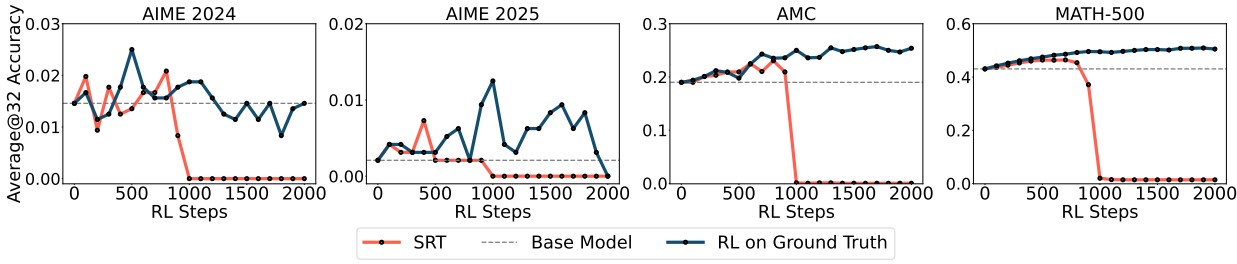

Figure 32: **(Training Deepseek-Math-7B-Instruct on MATH-12K using SRT)** We see similar trends for SRT-training on Deepseek-Math-7B-Instruct as we saw on our experiments with Qwen models: SRT initially matches performance gain obtained with RL training with ground truth, but faces performance collapse after prolonged training.

### K.2   Llama-3.1-8B-Instruct

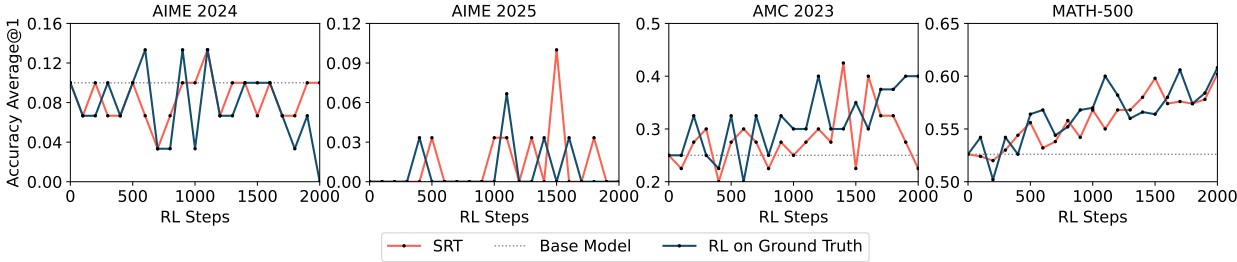

Figure 33: **(Training Llama-3.1-8B-Instruct on Big-Math-RL-Verified with learning rate $10^{-7}$)** Llama-3.1-8B-Instruct, when trained on a filter subset of the Big-Math dataset, shows significant gains on MATH-500 from both SRT and RL with ground truth. In fact, up to our training budget of 2K steps, both seem to improve performance at the same rate, from 52.6% to around 60%.

Llama-3.1-8B-Instruct showed no gains while being trained on DAPO or MATH-12K. This can be due to insufficient hyperparameter tuning or the model's starting performance on these datasets not suitable for learning. So we chose the Big-Math dataset (Albalak et al., 2025), a dataset with over 250,000 math questions with verifiable answers. Moreover, this dataset has been constructed by filtering common evaluation datasets like MATH-500, making it suitable for our purposes. **The primary benefit of using this dataset is that it comes with Llama-3.1-8B-Instruct pass rate**, so we can easily ascertain the difficulty of each datapoint and aggregate a subset that can be suitable for training the Llama model. Specifically, we take the subset of Big-Math where Llama-3.1-8B-Instruct has pass rate between 0.3 and 0.7, to filter away too

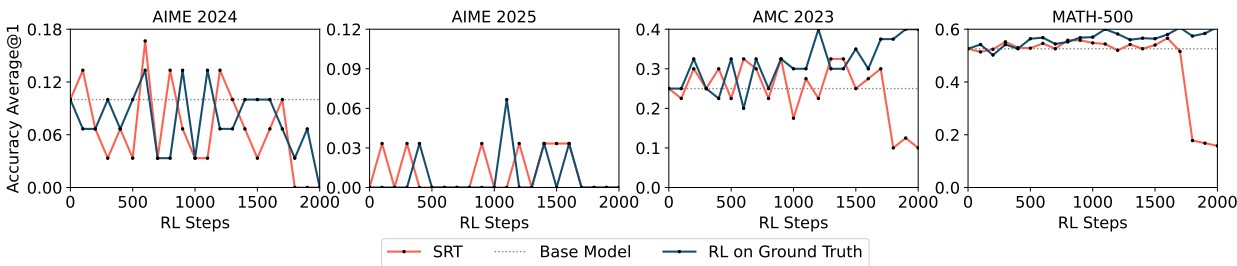

Figure 34: **(Training Llama-3.1-8B-Instruct on Big-Math-RL-Verified with learning rate** $3 \times 10^{-7}$**)** Llama-3.1-8B-Instruct, when trained on a filter subset of the Big-Math dataset with a higher learning rate ($3 \times 10^{-7}$ instead of $10^{-7}$ in Figure 33) demonstrates model collapse within the same training budget.

easy or too difficult questions. Next, we train the model on this subset using the same hyperparameters as we used for training the Qwen2.5-Math-7B-Instruct, except we lower the learning rate to $10^{-7}$ (from $10^{-6}$), as we see that leads to more stable learning curves. For evaluation, we use the same prompt template but temperature 0 (greedy decoding), to match the starting model's performance reported on its model card, and report pass@1 accuracy.

Figure 33 shows our results: on MATH-500, Llama-3.1-8B-Instruct shows the same performance growth when trained via SRT or RL with ground truth, up to our training budget of 2000 steps. Performance growth is also significant, and training improves pass@1 accuracy from 52.6% to around 60%. Note that we do note report performance on the harder datasets like AIME, because the Llama model's performance remain close to 0 throughtout training (with both objectives) on these datasets, signalling that they might be too hard for this model. We also ran an additional experiment using a higher learning rate of $3 \times 10^{-7}$. Figure 34 shows our empirical findings: with the higher learning rate, Llama-3.1-8B-Instruct also start to show model collapse within our training budget.

## L  Example Tasks from Reasoning Gym

**Task: Family Relationships (Level 4)**

*Question:* John is married to Isabella. They have a child called Edward. Edward is married to Victoria. What is Isabella to Edward? Respond only with the word that describes their relationship.

*Answer:* mother

**Task: Bitwise Arithmetic (Level 2)**

*Question:* Please solve this problem. Assume there is arbitrary bit depth and that there are signed integers. If the answer is negative, reply as a negative value (e.g., $-0x3$), not the two's-complement form. Reply only with the final hexadecimal value.

$$((0x3a24 - 0x24b8) + (0x1741 \gg 0x3))$$

*Answer:* 0x1854

**Task: Knights and Knaves (Level 2)**

*Question:* A very special island is inhabited only by sages and fools. Sages always tell the truth, and fools always lie. You meet 2 inhabitants: Zoey and Riley. Zoey commented, "Riley is a fool." In Riley's words: "Zoey is a sage or Riley is a sage." So who is a sage and who is a fool? (Format your answer like: "Zoey is a sage/fool, and Riley is a sage/fool")

*Answer:* Zoey is a fool, and Riley is a sage.

Above of we see one example from each of the three Reasoning Gym tasks used in our work. The examples shown are of the lowest difficulties that we first train the model on using ground truth RL. We do self training on more difficult variants on the tasks. Difficulty can be changed by the modifying the either number of person or digits. In this work, we abstract it away by calling it "level".

## M Chat Templates

**We use the default chat templates distributed with each model on Hugging Face.** VERL applies the model tokenizer's chat template before sending the resulting token sequence to vLLM; we do not define a custom template in our experiments. The training and evaluation chat templates were the same.

We do not provide a system prompt for any experiment except the BFCL tool-calling experiments mentioned in F. Nevertheless, two tokenizer families automatically insert system content even when the input dataset contains only a user message: (1) Qwen2.5-Math inserts its default mathematical reasoning instruction, and (2) Llama-3.1-Instruct inserts its standard system header, knowledge-cutoff line, and date line. The effective prompts used in our experiments are shown below.

**Qwen2.5-Math-7B.** The dataset supplies only the user message. The Hugging Face template inserts a prespecified system instruction.

```
<|im_start|>system
Please reason step by step, and put your final answer within \boxed{}.<|im_end|>
<|im_start|>user
{user prompt}<|im_end|>
<|im_start|>assistant
```

**Qwen3-4B-Base and Qwen3-14B-Base.** The dataset supplies only the user message, and the template does not insert a system message.

```
<|im_start|>user
{user prompt}<|im_end|>
<|im_start|>assistant
```

**Llama-3.1-8B-Instruct.** The dataset supplies only the user message. The Hugging Face template inserts the standard system header and metadata. We do not pass a `date_string` or otherwise override the template, so the template's default date, `26 Jul 2024`, is used.

```
<|begin_of_text|><|start_header_id|>system<|end_header_id|>

Cutting Knowledge Date: December 2023
Today Date: 26 Jul 2024

<|eot_id|><|start_header_id|>user<|end_header_id|>

{user prompt}<|eot_id|><|start_header_id|>assistant<|end_header_id|>
```

**DeepSeek-Math-7B-Instruct.** The dataset supplies only the user message, and the template does not insert a system message. Here `{bos_token}` denotes the model's beginning-of-sentence token.

```
{bos_token}User: {user prompt}

Assistant:
```

**Qwen3-4B for BFCL tool calling.** This is the only experiment in which the dataset explicitly supplies a system message. The BFCL system message contains the available function definitions and instructs the model to return Qwen-style tool-call JSON.

```
<|im_start|>system
# Tools
```

```
You may call one or more functions to assist with the user query.

You are provided with function signatures within <tools></tools> XML tags:
<tools>
{JSON function definitions}
</tools>

For each function call, return a json object with function name and arguments
within <tool_call></tool_call> XML tags:
<tool_call>
{"name": <function-name>, "arguments": <args-json-object>}
</tool_call>
Return only the tool call or tool calls.<|im_end|>
<|im_start|>user
{BFCL user prompt}<|im_end|>
<|im_start|>assistant
```

The expected model output is one or more blocks of the following form:

```
<tool_call>
{"name": "{function name}", "arguments": {argument JSON object}}
</tool_call>
```

