# OpenReview forum: "Can Large Reasoning Models Self-Train?"
_TMLR — Under review for TMLR_

### Review · Reviewer_bsEh · 2026-05-15

**Summary Of Contributions:**

This paper studies whether large reasoning models can self-train through online reinforcement learning without ground-truth labels, using majority voting over the model's own samples as a pseudo-reward. The authors evaluate this simple self-rewarded training (SRT) recipe on synthetic reasoning tasks and math reasoning benchmarks, comparing against RL with ground-truth rewards and several offline self-training baselines. The main positive finding is that SRT can improve held-out reasoning performance and the quality of majority-vote supervision in the early phase of training. The main negative finding is that prolonged self-rewarded RL can cause reward hacking and collapse, where the model learns prompt-independent template answers that maximize the pseudo-reward but hurt true accuracy. The paper also studies several mitigations, including early stopping, curriculum/easy-subset training, KL penalties, learning-rate changes, rollout-count changes, and entropy regularization.

Strengths:
- The paper contains a substantial empirical study across multiple models, datasets, RL algorithms, and training settings.
- The reward-hacking/model-collapse analysis is useful and clearly motivates why majority-vote self-training is not sufficient for sustained self-improvement.
- The comparison between fixed-teacher majority labels, online self-rewarding, and ground-truth RL helps clarify what is learned from the evolving policy.

Weaknesses:
- The core method is very close to TTRL and other concurrent majority-vote self-rewarded RL work; the paper's novelty is mainly empirical and diagnostic rather than algorithmic.
- Some reproducibility details and stability checks are missing, including chat templates, data shuffling seed sensitivity, and a no-KL ablation.

**Audience:**

Yes

**Audience Explanation:**

The findings should be interesting to researchers working on RLVR, self-training, test-time training, and reasoning models. In particular, the paper gives useful empirical evidence that majority-vote pseudo-rewards can be helpful in the short term but can fail badly under prolonged optimization. This is a relevant message for the community because many current self-improvement methods rely on self-consistency or model-generated feedback.

**Broader Impact Concerns:**

No.

**Claims And Evidence:**

Yes

**Claims Explanation:**

The empirical evidence supports the main claims that majority-vote self-rewarded RL can improve early training performance and that prolonged training can lead to reward hacking and collapse. The collapse analysis is especially convincing because the paper tracks held-out accuracy, pseudo-reward, KL divergence, entropy, and qualitative model outputs.

However, I think the paper should be more precise about the scope of its claims. The evidence mainly supports improvements in pass@1/average-sample accuracy and majority-vote accuracy. It does not clearly establish improvement in pass@k for large k, where diversity effects may matter. Also, because the method is very similar to TTRL's majority-vote reward construction, the paper should avoid presenting SRT as a substantially new algorithm and instead emphasize its broader held-out evaluation, collapse analysis, and mitigation study.

**Requested Changes:**

Critical to my recommendation:

1. Clarify novelty and relationship to TTRL. TTRL uses majority voting over model samples to construct pseudo-rewards for RL on unlabeled data, which is essentially the same core reward mechanism as SRT. Please make the distinction from TTRL more explicit in the introduction and related work. In my view, the main contribution here is not a new reward mechanism, but a broader empirical study of held-out self-training, reward hacking, collapse, and mitigation strategies.

2. Standardize and define the evaluation metrics. The paper uses pass@1, avg@k, Average@32, Majority@32, maj@k, and sometimes different metrics across figures. Please add a compact metric-definition paragraph and make the figure captions consistent. In particular, for Llama-3.1-8B-Instruct under greedy decoding, Majority@32 is equivalent to Average@32/pass@1, so labeling it as Majority@32 can be misleading without a very explicit note.

3. Improve reproducibility details. Please provide the exact chat/prompt templates for each model and specify whether the same templates are used for training and evaluation.

Important but not necessarily critical:

4. Add a seed-stability check, at least for Qwen2.5-Math-7B on a representative dataset. Because the paper studies peak performance and collapse timing, it would be helpful to know whether these curves are stable under different training data shuffles or random seeds.

5. Add a no-KL ablation or explain why it is omitted. Appendix G studies several KL coefficients, but I did not see KL coefficient 0. This would help isolate whether KL regularization changes the onset or form of collapse.

6. Rephrase the discussion around Eq. 6. Eq. 6 is the binary pseudo-reward induced by agreement with the majority answer, not the full SRT objective. Calling it the "SRT objective" is slightly imprecise.

7. Clarify Table 2 in Appendix F. The "AMC/AIME" column appears to aggregate AMC, AIME 2024, and AIME 2025. Please rename this column or explicitly state the exact averaging procedure.

---

> ### Author Response · Authors · 2026-07-20
> **Response to Reviewer bsEh**
>
> We thank Reviewer bsEh for their constructive comments and thoughtful review of our work. We have addressed their concerns below, and would love to engage with them to answer any remaining questions or comments.
>
> **For the comments mentioned under Requested Changes as critical to the reviewer’s recommendation**:
>
> > Clarify novelty and relationship to TTRL
>
> We thank the reviewer for highlighting the close relationship between SRT and TTRL. We agree that both methods use majority voting over multiple model-generated solutions to construct pseudo-rewards for reinforcement learning on unlabeled problems. We will revise the introduction and related-work sections to make this overlap explicit and avoid presenting the majority-vote reward mechanism itself as the primary contribution of SRT.
>
> However, as the reviewer noted, **our intended contribution is broader than proposing this reward construction**.
>
> 1. Firstly, despite the similarities in the training objective, the fundamental object of study is different between TTRL and SRT. **TTRL studies majority-vote-based reinforcement learning primarily as a test-time adaptation method**, where the model is updated directly on an unlabeled evaluation set. In contrast, **SRT investigates the same general self-rewarding principle as a held-out self-training procedure**: the model is trained on unlabeled problems and evaluated on separate held-out tasks. This setting allows us to study whether improvements reflect transferable reasoning gains rather than adaptation to a fixed test set.
>
> 2. **We also study the mechanistic feasibility of such a training pipeline under a diverse set of training conditions**, describe the resulting reward hacking and study the potential mitigation strategies, all of which constitute a complementary study to concurrent works such as TTRL.
>
> We will accentuate these differences in the camera ready version of the manuscript.
>
> > Standardize and define the evaluation metrics
>
> Thanks for pointing this out. We have added a paragraph under Section 4.1 in the updated manuscript that defines our metrics. In all the relevant figures we now use “Accuracy Average @k” or “Average Accuracy @k”, both of which are equivalent. In a few figures where it is deemed ambiguous we have added the information in the caption.
>
> > Improve reproducibility details. Please provide the exact chat/prompt templates for each model and specify whether the same templates are used for training and evaluation.
>
> We thank the reviewer for catching this. We have added all the chat templates alongside their system prompts in Appendix M in the revised version of the paper.
>
> **Next, we also address below the non-critical changes requested by the reviewer:**
>
> > Add a seed-stability check, at least for Qwen2.5-Math-7B on a representative dataset.
>
> Thanks for this suggestion. We added a dataset shuffling experiment for training the Qwen2.5-Math-7B model on the DAPO dataset. The results can be found in Appendix H.6, and we have also included the summary figure in [this anonymous link](https://ibb.co/ccFj58zD) for the reviewer’s convenience. **According to our experiments, the model behavior is fairly stable across all the 4 seeds that we have experimented with**.
>
> > Add a no-KL ablation or explain why it is omitted.
>
> This is a good suggestion! We added a no KL experiment in Figure 22 in Appendix H.2 in the updated manuscript. Setting the KL coefficient to 0 does not significantly change the model behavior, as it still results in reward hacking and performance collapse. The figure can also be found in [this anonymous link](https://ibb.co/SXdb7nbf).
>
> > Rephrase the discussion around Eq. 6.
>
> Thanks for pointing this out, and we have rephrased the discussion around Equation 6 in our updated manuscript.
>
> > Clarify Table 2 in Appendix F
>
> We have fixed the issue in the updated manuscript.
>
> **We thank the reviewer again, and look forward to engaging with them to address any remaining concerns about our manuscript**.

---

### Review · Reviewer_GAbb · 2026-05-18

**Summary Of Contributions:**

**Summary**: This paper studies self-training for large language models (LLMs) within reinforcement learning (RL). Specifically, the authors introduce Self-Rewarded Training (SRT). This method leverages majority voting across multiple model-generated outputs as a pseudo-label (self-supervision signal) for RL during post-training. They find that SRT can lead to measurable performance gains on reasoning benchmarks and simultaneously improve the learning signal itself. They also identify a critical limitation of training with self-generated rewards, reward hacking, and propose mitigations such as early stopping and curriculum learning.

**Strength**:

- The idea of using majority voting as a self-supervision signal for RL training of LLMs is simple yet effective.
- The presentation is clear and well-structured. I personally find the paper enjoyable to read.
- The experiments are comprehensive, and the analyses are thorough. The analysis of model collapse provides valuable insights.

**Weakness**:
- The real-world experiments are restricted to math. Validating the method on other domains where majority voting is feasible, would strengthen the paper.

**Additional Comments:**

NA

**Audience:**

Yes

**Audience Explanation:**

This paper will be of interest to researchers working on RL for LLM training.

**Claims And Evidence:**

Yes

**Claims Explanation:**

- SRT improves an LLM's reasoning abilities beyond the base model's capabilities. Supported by Figures 2, 3, 4, and 12–17.
- The evolving teacher beats the fixed teacher is supported by Figure 2.
- Prolonged SRT causes collapses. Figures 6, 7, 12–17, and 29 show collapse across all models. The training-dynamics plots in Figure 7 (sudden KL spike co-occurring with reward saturation and accuracy collapse) make the diagnosis particularly compelling.
- Early stopping and easy-subset curriculum mitigate collapse. Supported by Figures 9, 10, 26, 27, and 28.
- Some of the other methods, such as increasing the KL coefficient or reducing learning, do not help to mitigate model colapse. This claim is supported by Figures 20 and 21.

**Requested Changes:**

Could the authors extend the experiments to at least one non-math domain where majority voting is well-defined? Even a small experiment here would substantially strengthen the generality claim of the paper.

---

> ### Author Response · Authors · 2026-07-20
> **Response to Reviewer GAbb**
>
> We thank Reviewer GAbb for their time to review our work and the thoughtful suggestions, we believe they will greatly strengthen our paper. We address their requested changes below.
>
> > Could the authors extend the experiments to at least one non-math domain where majority voting is well-defined? Even a small experiment here would substantially strengthen the generality claim of the paper.
>
> As an example of self-training in a non-math domain, we added an experiment using tool calling where there is still a well defined notion of the majority vote. The main results are in Appendix F and Figure 21 of the updated manuscript, which you can also find [in this anonymous link](https://ibb.co/Y7S7LMRp).
>
> **Below, we add a short description of our experiment for the reviewer's convenience**.
>
> (**Task description**) We used the single-turn AST component of the Berkeley Function Calling Leaderboard (BFCL) taskset [1]. For training, we use all 1,000 Python examples from BFCL V1: 400 simple-function, 200 multiple-function, 200 parallel-function, and 200 parallel-multiple examples. For evaluation, we use 240 held-out examples from the newer, real-world BFCL V2 Live distribution: 100 live_simple, 100 live_multiple, 16 live_parallel, and 24 live_parallel_multiple examples. Each evaluation result is averaged over 32 sampled responses.
>
> (**Model**) We train a Qwen3-4B model with default thinking mode on, which already possesses substantial tool-calling capability and we show that SRT can further boost the tool calling performance of this model, on par with RL with ground truth supervision.
>
> (**Summary Results**) A summary of Figure 21 from the updated manuscript is given below where we report the best score of each of the training method (SRT and RL with Ground Truth):
>
> | BFCL Live dataset | Base Qwen3-4B | RL with gold reward | SRT |
> | :--- | :---: | :---: | :---: |
> | Simple | 84.4% | 86.6% | 85.9% |
> | Multiple | 84.9% | 87.8% | 89.1% |
> | Parallel | 72.8% | 77.9% | 77.1% |
> | Parallel multiple | 74.8% | 87.6% | 88.8% |
> | **Overall average** | **79.2%** | **85.0%** | **85.2%** |
>
> We are happy to address any other concerns that the reviewer may have about our manuscript.
>
> ## References
>
> [1] Shishir G Patil, Huanzhi Mao, Fanjia Yan, Charlie Cheng-Jie Ji, Vishnu Suresh, Ion Stoica, Joseph E. Gonzalez. The Berkeley Function Calling Leaderboard (BFCL): From Tool Use to Agentic Evaluation of Large Language Models, Proceedings of the 42nd International Conference on Machine Learning, PMLR 267:48371-48392, 2025

---

### Review · Reviewer_tSj5 · 2026-07-06

**Summary Of Contributions:**

This paper studies the impact of self-training for training reasoning language models within a reinforcement learning setting, specifically the authors make use of majority voting as the direct self-feedback mechanism and evaluate it across a comprehensive suite of synthetic and real reasoning tasks.

The study shows that this straightforward self-training method enhances both the model's reasoning performance and its ability to generate high-quality feedback for subsequent RL iterations. Besides, they identify some significant limitations: prolonging the RL training depending on self-reward could lead to 'reward hacking', thus causing the model to manage to maximize pseudo-rewards and inducing a sudden performance collapse.

They finally conclude by emphasizing that enhancing the self-improvement of RL training for LLMs requires more robust feedback design.

**Audience:**

Yes

**Audience Explanation:**

1. The application of reinforcement learning to automated reasoning is an established, albeit specialized, area of research. While not broadly critical to the entire TMLR audience, there is a dedicated subset of the community that tracks developments within this specific niche.

2. The paper's empirical documentation of "reward hacking" during self-training offers an acceptable observation. While this is a known challenge for autonomous self-improvement, the specific findings presented here will likely only resonate with researchers directly focused on this narrow bottleneck, making it a relatively modest contribution to the field.

**Broader Impact Concerns:**

There are no obvious malicious ethical implications inherent to the training methodology itself. However, deploying autonomous reasoning models that are susceptible to sudden, unprompted "performance collapse" due to reward hacking could pose severe reliability and safety risks in real-world applications.

**Claims And Evidence:**

Yes

**Claims Explanation:**

1. The paper thoroughly details these experimental setups, the empirical demonstration of both the initial performance gains and the subsequent reward hacking provides a clear, evidence-based foundation for their conclusions.

2. Even if the majority voting mechanism itself is narrow, documenting the exact mechanics of "reward hacking" and the sudden performance collapse in online self-training provides a cautionary case study.

3. The paper's comparative analysis across both synthetic and real reasoning data highlights how feedback dynamics diverge when moving away from perfectly controlled environments. This aspect of the study is useful for researchers focused on evaluation design and benchmarking, independently of whether they use the specific self-training protocol proposed.

**Requested Changes:**

1. Please provide a granular analysis or ablation study identifying the exact tipping point where the model transitions from constructive self-improvement to reward hacking.

2. While the abstract rightly identifies feedback design as the central challenge moving forward, the submission would be significantly strengthened by proposing or testing preliminary mitigation strategies (e.g., early stopping, regularization, or dynamic thresholding) to delay or prevent the sudden performance collapse.

3. Ensure that the differences in reward hacking onset between the "synthetic" and "real" reasoning tasks are clearly contrasted in the results section.

---

> ### Author Response · Authors · 2026-07-20
> **Responses to Reviewer tSj5**
>
> We thank Reviewer tSj5 for their positive review and thoughtful suggestions. We address their requested changes below, and would love to engage the reviewer to address any remaining concerns.
>
> > Please provide a granular analysis or ablation study identifying the exact tipping point where the model transitions from constructive self-improvement to reward hacking.
>
> Thank you for this valuable suggestion. We did a granular analysis of the model transitions by saving model generated responses in every step of the training. Our key observations are listed below:
>
> 1. Qwen2.5-Math-7B, when self trained on the DAPO dataset, completely collapses at about step 620 with generations like “\\boxed{1}”. However, even before the final model collapse happens, there are signs of reward hacking in the model generations. For example, **before the model collapses into \\boxed{1}, there is a phase where the model produces long plausible looking chains-of-thoughts, but starts to produce the final answer as \\boxed{1}**.
>
> 2. For the DAPO trained model during steps 1–300: roughly 50\% of responses are long, but only 3–4\% of either long or short responses answer \\boxed{1}. However, at around steps 301–500: \\boxed{1} becomes more common among long responses. At step 446, 55.5\% of responses remain long, and 28.5\% of those long responses end in \\boxed{1}, versus only 3.9\% of short responses. Around steps 501–600, the fake-long-CoT phenomenon becomes strongest. At step 577, 39.3\% of responses are long and 44.8\% of those end in \\boxed{1}.
>
> 3. However, at about step 600, the model stops generating chains-of-thought entirely and just produces \\boxed{1} (and some additional meaningless tokens) as the final response.
>
> We added this observation and analysis in Section 4.3 of the updated manuscript. We also attach an [anonymous link](https://ibb.co/pBT7J840) to the main granular analysis figure here for the reviewer’s convenience.
>
> > While the abstract rightly identifies feedback design as the central challenge moving forward, the submission would be significantly strengthened by proposing or testing preliminary mitigation strategies (e.g., early stopping, regularization, or dynamic thresholding) to delay or prevent the sudden performance collapse.
>
> In our original manuscript, we have already experimented with several mitigation strategies for reward collapse in Section 5. Specifically, we show that Early stopping with a small labeled validation dataset can effectively identify the model collapse (Section 5.1). We also show that some forms of curriculum learning can delay model collapse up to the end of our training budget (Section 5.2). Additionally, we experimented with additional intervention mechanisms such as tuning the KL coefficient to increase regularization, or tuning the learning rate/number of generations per prompt, but many of the intuitive changes did not result in solving the reward hacking problem. These later experiments are included in appendix H in the new manuscript (Appendix G in the original one).
>
> > Ensure that the differences in reward hacking onset between the "synthetic" and "real" reasoning tasks are clearly contrasted in the results section.
>
> The nature of reward hacking in the synthetic examples of reasoning gym is subtly different from reward hacking in math domain. For example, in a difficult Knights and Knave dataset, the model reward hacks into producing not a single universal answer such as \boxed{1} like in the case of math. Instead, the model learns to label every person as the truth-telling class. The class name varies by prompt: hero, saint, knight, sage, pioneer, angel, altruist, etc. Consequently, the degenerate answer string is different for every prompt. Examples include:
>
> 1. “\boxed{Grace is a hero, Samuel is a hero, and Joseph is a hero}”,
>
> 2. “\boxed{Ava is a sage, Sophia is a sage, Henry is a sage, …}”
>
> The nature of this kind of reward hacking is largely due to the fact that the reasoning gym evaluator requires the model to output the names given in the question in order to be assigned a valid score and a simple \\boxed{one word} wouldn’t qualify as an answer. **In short, SRT still reward hacks on synthetic tasks, but the particular form of the reward hacking solution depends on the evaluation function of the specific task at hand**.
>
> Moreover, the reward hacking dynamic is also somewhat different in synthetic Knights and Knave where the unique answer per prompt increases first before decreasing while for math tasks, it decreases uniformly. We have added this analysis in the updated manuscript in Section 4.4 and Figure 9 in the updated manuscript shows the granular trajectory of model output length and pseudo reward comparison for Reasoning Gym and DAPO. Here is an [anonymous link](https://ibb.co/pBT7J840) for the same figure.
>
> **Please let us know if there are any additional concerns that we can address.**